# Learning from higher-order correlations, efficiently: hypothesis tests, random features, and neural networks

**Eszter Székely**[†] [*]      **Lorenzo Bardone** [†]      **Federica Gerace**[‡]

**Sebastian Goldt**[§]

International School of Advanced Studies (SISSA)
Trieste, Italy

## Abstract

Neural networks excel at discovering statistical patterns in high-dimensional data sets. In practice, higher-order cumulants, which quantify the non-Gaussian correlations between three or more variables, are particularly important for the performance of neural networks. But how efficient are neural networks at extracting features from higher-order cumulants? We study this question in the spiked cumulant model, where the statistician needs to recover a privileged direction or "spike" from the order-$p \geq 4$ cumulants of $d$-dimensional inputs. We first discuss the fundamental statistical and computational limits of recovering the spike by analysing the number of samples $n$ required to strongly distinguish between inputs from the spiked cumulant model and isotropic Gaussian inputs. Existing literature established the presence of a wide statistical-to-computational gap in this problem. We deepen this line of work by finding an exact formula for the likelihood ratio norm which proves that statistical distinguishability requires $n \gtrsim d$ samples, while distinguishing the two distributions in polynomial time requires $n \gtrsim d^2$ samples for a wide class of algorithms, i.e. those covered by the low-degree conjecture. Numerical experiments show that neural networks do indeed learn to distinguish the two distributions with quadratic sample complexity, while "lazy" methods like random features are not better than random guessing in this regime. Our results show that neural networks extract information from higher-order correlations in the spiked cumulant model efficiently, and reveal a large gap in the amount of data required by neural networks and random features to learn from higher-order cumulants.

## 1 Introduction

Discovering statistical patterns in high-dimensional data sets is the key objective of machine learning. In a classification task, the differences between inputs in different classes arise at different statistical levels of the inputs: two different classes of images will usually have different means, different covariances, and different higher-order cumulants (HOCs), which describe the non-Gaussian part of the correlations between three or more pixels. While differences in the mean and covariance allow for rudimentary classification, Refinetti *et al.* [1] recently highlighted the importance of HOCs for the performance of neural networks: when they removed the HOCs per class of the CIFAR10 training

---

[*]Current address: CCFE, Culham Science Centre, Abingdon, Oxon, OX14 3DB, UK

[†]These authors contributed equally.

[‡]Current address: Dipartimento di Matematica, Universita' di Bologna, Bologna (BO), Italy

[§]Correspondence to: {eszekely, lbardone, fgerace, sgoldt}@sissa.it

38th Conference on Neural Information Processing Systems (NeurIPS 2024).

set, the test accuracy of various deep neural networks dropped by up to 65 percentage points. The importance of higher-order cumulants (HOCs) for classification in general and the performance of neural networks in particular raises some fundamental questions: what are the fundamental limits of learning from HOCs, i.e. how many samples $n$ ("sample complexity") are required to extract information from the HOCs of a data set? How many samples are required when using a *tractable* algorithm? And how do neural networks and other machine learning methods like random features compare to those fundamental limits?

In this paper, we study these questions by analysing a series of binary classification tasks. In one class, inputs $x \in \mathbb{R}^d$ are drawn from a normal distribution with zero mean and identity covariance. These inputs are therefore isotropic: the distribution of the high-dimensional points projected along a unit vector in any direction in $\mathbb{R}^d$ is a standard Gaussian distribution. Furthermore, all the higher-order cumulants (of order $p \geq 3$) of the inputs are identically zero. Inputs in the second class are also isotropic, except for one special direction $u \in \mathbb{R}^d$ in which their distribution is different. This direction $u$ is often called a "spike", and it can be encoded in different cumulants: for example, we could "spike" the covariance by drawing inputs from a Gaussian with mean zero and covariance $\mathbb{1} + \beta u u^\top$; the signal-to-noise ratio $\beta > 0$ would then control the variance of $\lambda = \langle u, x \rangle$. Likewise, we could spike a higher-order cumulant of the distribution and ask: what is the minimal number of samples required for a neural network trained with SGD to distinguish between the two input classes? This simple task serves as a proxy for more generic tasks: a neural network cannot be able to *extract* information from a given cumulant if it cannot even recognise that it is different from an isotropic one.

We can obtain the fundamental limits of detecting spikes at different levels of the cumulant hierarchy by considering the hypothesis test between a null hypothesis (the isotropic multivariate normal distribution with zero mean and identity covariance) and an alternative "spiked" distribution. We can then compare the sample complexity of neural networks to the number of samples necessary to distinguish the two distributions using unlimited computational power, or efficiently using algorithms that run in polynomial time. A second natural comparison for neural networks are random features or kernel methods. Since the discovery of the neural tangent kernel [2], and the practical success of kernels derived from neural networks [3, 4], there has been intense interest in establishing the advantage of neural networks with *learnt* feature maps over classical methods with *fixed* feature maps, like kernel machines. The role of higher-order correlations for the relative advantage of these methods has not been studied yet.

In the following, we first introduce some fundamental notions around hypothesis tests and in particular the low-degree method [5–8], which will be a key tool in our analysis, using the classic spiked Wishart model for sample covariance matrices [9, 10]. For spiked cumulants, the existing literature (see section 1.1 for a complete discussion) establishes the presence of a wide statistical-to-computational gap in this model. Our **main contributions** are then as follows:

- We deepen the understanding of the statistical-to-computational gap for learning from higher-order correlations on the statistical side by showing that at *unbounded computational power*, a number of samples *linear* in the input dimension is required to reach *statistical distinguishability*. We prove this by explicitly computing the norm of the *likelihood ratio*, see theorem 2.

- On the algorithmic side, SQ bounds and previous low-degree analyses [11, 12], showed that the sample complexity of learning from HOCs is instead *quadratic* for a wide class of polynomial-time algorithms (section 3.2). Here, we provide a different, more direct proof of such a bound.

- Using these fundamental limits on learning from HOCs as benchmarks, we show numerically that neural networks learn the mixed cumulant model efficiently, while random features do not, revealing a large separation between the two methods (sections 4.1 and 4.2).

- We finally show numerically that the distinguishability in a simple model for images [13] is precipitated by a cross-over in the behaviour of the higher-order cumulants of the inputs (section 4.3).

## 1.1 Further related work

**Detecting spikes in high-dimensional data**    There is an enormous literature on statistical-to-computational gaps in the detection and estimation of variously structured principal components of high-dimensional data sets. This problem has been studied for the Wigner and Wishart ensembles of random matrices [9, 14–24] and for spiked tensors [25–33]. For comprehensive reviews of the topic, see refs. [34–37]. While the samples in these models are often non-Gaussian, depending on the prior distribution over the spike $u$, the spike appears already at the level of the covariance of the inputs. Here, we instead study a high-dimensional data model akin to the one used by Wang & Lu [38] to study online independent component analysis (ICA). This data model can be interpreted as having an additional whitening step to pre-process the inputs, which is a common pre-processing step in ICA [39], hence inputs have identity covariance even when cumulants of order $p \geq 3$ are spiked. Wang & Lu [38] proved the existence of a scaling limit for online ICA, but did not consider the sample complexity of recovering the spike/distinguishing the distributions, which is the focus of this paper.

**NGCA, Gaussian pancakes and low-degree polynomials**    Related models have been introduced under the name of *Non-Gaussian Component Analysis* (NGCA) [40], and studied from the point of view of *statistical query* (SQ) complexity in a sequence of papers [41–46]. In particular [11] nicknames *Gaussian pancakes* a class of models that includes the *spiked cumulant model* presented here. The SQ bounds found in [11] and refined in the subsequent works (see Diakonikolas *et al.* [47] and references therein) could be used, together with SQ-LDLR equivalence [48], to provide estimates on low-degree polynomials. Finally, [12] proves very general bounds on LDLR norm; our theorem 5 provides an alternative derivation of these bounds, in a setting that is closer to the experiments in section 4.

**Separation between neural networks and random features**    The discovery of the neural tangent kernel by Jacot *et al.* [2] and the flurry of results on linearised neural networks [2, 3, 49–55] has triggered a new wave of interest in the differences between what neural networks and kernel methods can learn efficiently. While statistical separation results have been well-known for a long time [56], recent work focused on understanding the differences between random features and neural networks *trained by gradient descent* both with wide hidden layers [4, 57–61] or even with just a few hidden neurons [62, 63]. At the heart of the data models in all of these theoretical models is a hidden, low-dimensional structure in the task, either in input space (for mixture classification) or in the form of single- or many-index target functions. The impact of higher-order input correlations in mixture classification tasks on the separation between random features and neural networks has not been directly studied yet.

**Reproducibility**    We provide code for all of our experiments, including routines to generate the various synthetic data sets we discuss, on GitHub https://github.com/eszter137/data-driven-separation.

## 2   The data models

Throughout the paper, we consider binary classification tasks where high-dimensional inputs $x^\mu = (x_i^\mu) \in \mathbb{R}^d$ have labels $y^\mu = \pm 1$. The total number of training samples is $2n$, i.e. we have $n$ samples per class. For the class $y^\mu = -1$, inputs $x^\mu = z^\mu$, where $z^\mu \sim_{\mathrm{iid}} \mathcal{N}(0, \mathbb{1}_d)$ and $\mathcal{N}$ denotes the normal distribution. For the class $y^\mu = 1$ instead, we consider "spiked" input models as follows.

**The Gaussian case**    The simplest spiked model is the spiked Wishart model from random matrix theory [9, 10], in which

$$x^\mu = \sqrt{\frac{\beta}{d}} g^\mu u + z^\mu, \quad g^\mu \sim_{\mathrm{iid}} \mathcal{N}(0, 1), \tag{1}$$

where $u = (u_i)$ is a $d$-dimensional vector with norm $\|u\| = \sqrt{d}$ whose elements are drawn element-wise i.i.d. according to some probability distribution $\mathcal{P}_u$. In this model, inputs are Gaussian and indistinguishable from white noise except in the direction of the "spike" $u$, where they have variance $1 + \beta$, where $\beta > 0$ is the *signal-to-noise ratio*. By construction, all higher-order cumulants are zero.

**The spiked cumulant model** To study the impact of HOCs, we draw inputs in the "spiked" class from the data model used by Wang & Lu [38] to study online independent component analysis (ICA). First, we sample inputs

$$\tilde{x}^\mu = \sqrt{\frac{\beta}{d}} g^\mu u + z^\mu, \qquad g^\mu \sim_{\text{i.i.d.}} p_g, \tag{2}$$

as in the Wishart model, but crucially sample the latent variables $g^\mu$ from some non-Gaussian distribution $p_g(g^\mu)$, say, the Rademacher distribution; see assumption 1 for a precise statement. For any non-Gaussian $p_g$, the resulting inputs $\tilde{x}^\mu$ have a non-trivial fourth-order cumulant proportional to $\kappa_4^g u^{\otimes 4}$, where $\kappa_4^g \equiv \mathbb{E}(g^\mu)^4 - 3\mathbb{E}(g^\mu)^2$ is the fourth cumulant of the distribution of $g^\mu$. We fix the mean and variance of $p_g$ to be zero and one, respectively, so the covariance of inputs has a covariance matrix $\Sigma = \mathbb{1}_d + \beta/d\, uu^\top$. To avoid trivial detection of the spike from the covariance, the key ingredient of the spiked cumulant model is that we whiten the inputs in that class, so that the inputs are finally given by

$$x^\mu = S\tilde{x}^\mu, \qquad S = \mathbb{1} - \frac{\beta}{1 + \beta + \sqrt{1+\beta}} \frac{uu^\top}{d}, \tag{3}$$

with the whitening matrix $S$ (see appendix B.4.3). Hence inputs $x^\mu$ are isotropic Gaussians in all directions except $u$, where they are a weighted sum of $g^\mu$ and $\langle z, u \rangle$. The whitening therefore changes the interpretation of $\beta$: rather than being a signal-to-noise ratio, as in the spiked Wishart model, here $\beta$ controls the quadratic interpolation between the distributions of $g^\mu$ and $z$ in the direction of $u$ (see eq. (55) in the appendix). This leaves us with a data set where inputs in both classes have an average covariance matrix that is the identity, which means that PCA or linear neural networks [64–67] cannot detect any difference between the two classes.

## 3 How many samples do we need to learn?

Given a data set sampled from the spiked cumulant model, we can now ask: how many samples does a statistician need to reliably detect whether inputs are Gaussian or not, i.e. whether HOCs are spiked or not? This is equivalent to the hypothesis test between $n$ i.i.d. samples of the isotropic normal distribution in $\mathbb{R}^d$ as the null hypothesis $\mathbb{Q}_{n,d}$, and $n$ i.i.d. samples of the spiked cumulant model eq. (3) as the alternative hypothesis $\mathbb{P}_{n,d}$. In section 3.1 we will first consider the problem from a statistical point of view, assuming to have *unbounded computational power* and no restrictions on the distinguishing algorithms. Then, in section 3.2 we will use the *low-degree method* to understand how the picture changes when we restrict to algorithms whose running time is at most polynomial in the space dimension $d$.

### 3.1 Statistical distinguishability: LR analysis

Recall the notion of *strong asymptotic distinguishability*: two sequences of probability measures are strongly distinguishable if it is possible to design statistical tests that can classify correctly which of the two distributions a sample was drawn from with probabilities of type I and II errors that converge to 0 (see appendix B.2 for the precise definition). Using this definition of distinguishability, we will ask what is the *statistical sample complexity exponent*, i.e. the minimum $\theta$ such that in the high-dimensional limit $d \to \infty$, if $n \asymp d^\theta$, then $\mathbb{P}_{n,d}$ and $\mathbb{Q}_{n,d}$ are strongly distinguishable (with no constraints on the complexity of statistical tests).

A useful quantity to consider is the *likelihood ratio* (LR) of probability measures, which is defined as

$$L(x) := \frac{d\mathbb{P}}{d\mathbb{Q}}(x). \tag{4}$$

Computing the LR norm $||L||^2 := \mathbb{E}_\mathbb{Q}[L^2]$ is an excellent tool to probe for distinguishability: if $(\mathbb{P}_{n,d})$ and $(\mathbb{Q}_{n,d})$ are strongly distinguishable, then $||L_{n,d}|| \to \infty$ (this is the well-known *second moment method for distinguishability*, see proposition 6 in the appendix for the precise statement). In the following we will apply this method, finding a formula for the LR norm and then study its limit as a function of $\theta$. Here and throughout, we will denote the data matrix by $\underline{x} = (x^\mu)_{1,\dots,n}^\top$; in general matrices of size $n \times d$ will be denoted with underlined letters; see appendix B.1 for a complete summary of our notation. We will use Hermite polynomials, denoted by $(h_k)_k$, see appendix B.3

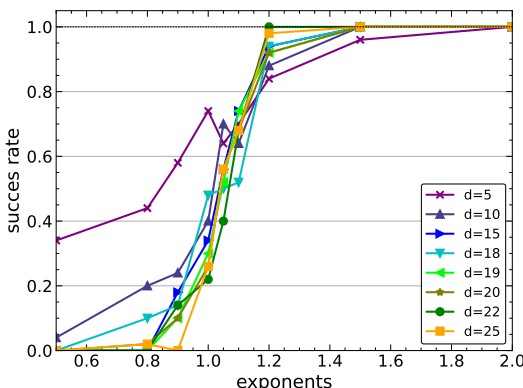

Figure 1: **The performance of an exhaustive-search algorithm corroborates the presence of a phase transition for $\theta = 1$, as suggested by theorem 2.** Success rate of an exponential-time search algorithm over all the possible spikes in the $d$-hypercube as a function of the exponent $\theta$ that quantifies as $n = d^\theta$ the samples used in the log-likelihood test (8), in the $g \sim \text{Radem}(1/2)$ case.

for details. We assume that the spike $u$ is drawn from a known prior $\mathcal{P}(u)$, and that the scalar latent variables $(g^\mu)_{\mu=1,\dots,n}$ are drawn i.i.d. from a distribution $p_g$ with the following properties:

**Assumption 1** (Assumption on latent variables $g^\mu$). *We assume that the one-dimensional probability distribution of the non-Gaussian random variable $p_g(g)$ is an even distribution, $p_g(g = dx) = p_g(-g = -dx)$, with mean 0 and variance 1, and that it satisfies the following requirement on the growth of its Hermite coefficients:*

$$\mathbb{E}\left[h_m(g)\right] \leq \Lambda^m m! \tag{5}$$

*where $\Lambda > 0$ is a positive constant that does not depend on $m$. Finally, we assume that $p_g$ has tails that cannot go to 0 slower than a standard Gaussian, $\mathbb{E}[\exp(g^2/2)] < +\infty$.*

A detailed discussion of these assumptions can be found in appendix B.4, as well as a proof that they are satisfied by a wide class of distributions including all the compactly supported distributions with mean 0 and variance 1 (some concrete examples are $p_g = \text{Rademacher}(1/2)$ or $p_g = \text{Unif}(-\sqrt{3}, \sqrt{3})$).

Under these assumption we can compute the following formula for the LR norm.

**Theorem 2.** *Suppose that $u$ has i.i.d. Rademacher prior and that the non-Gaussian distribution $p_g$ satisfies assumption 1. Then the norm of the total LR is given by*

$$\|L_{n,d}\|^2 = \sum_{j=0}^{d} \binom{d}{j} \frac{1}{2^d} f\left(\beta, \frac{2j}{d} - 1\right)^n, \tag{6}$$

*where $f$ is defined as the following average over two independent replicas $g_u, g_v \sim g$ of $g$:*

$$f(\beta, \lambda) := \mathop{\mathbb{E}}_{g_u, g_v} \left[ \frac{1 + \beta}{\sqrt{(1+\beta)^2 - \beta^2 \lambda^2}} e^{-\frac{(1+\beta)\left((1+\beta)(g_u^2 + g_v^2) - 2\beta(g_u g_v)\lambda\right)}{2(1+\beta)^2 - 2\beta^2\lambda^2} + \frac{g_u^2 + g_v^2}{2}} \right] \tag{7}$$

We prove theorem 2 in appendix B.5. The theorem has key consequences in two directions:

- on the one hand, eq. (6) implies that the LR norm is bounded for $\theta < 1$ (see lemma 13 in appendix B.5.3), which confirms that below that threshold *strong distinguishability is impossible*;

- on the other hand, eq. (6) implies that whenever there exists $\tilde{\lambda}$ such that $f(\beta, \tilde{\lambda}) > 1$, we find that $\|L_{n,d}\| \geq \frac{f(\beta, \tilde{\lambda})^n}{2^d}$. Thus, the LR norm diverges as soon as $n$ grows as $n \asymp d^\theta$ with any $\theta > 1$. In appendix B.5.3 we detail as an example the case in which $g \sim \text{Radem}(1/2)$ where the norm $\|L_{n,d}\|$ even diverges for $\theta = 1$ and $d \asymp \gamma n$ for some $\gamma > 0$.

So, besides the intrinsic value of providing an exact formula for the LR of the spiked cumulant model, theorem 2 implies the presence of a phase transition at $\theta = 1$ for the *strong statistical distinguishability*.

A complementary approach that substantiates the presence of the statistical phase transition at $\theta = 1$ can be seen in fig. 1, where we perform a maximum log-likelihood test along $u \cdot x^\mu$ for *all* the possible spikes $u$ in the $d$-dimensional hypercube using the formula for the LR conditioned on the spike,

$$\sum_{\mu=1}^{n} \log \left( \frac{p_x(x^\mu | u)}{p_z(x^\mu)} \right) = \sum_{\mu=1}^{n} \log \mathbb{E}_g \sqrt{1+\beta} \exp \left( -\frac{1+\beta}{2} \left( g - \sqrt{\frac{\beta}{(1+\beta)d}} x^\mu \cdot u \right)^2 + \frac{g^2}{2} \right),$$

(8)

(see eq. (61) in appendix B.5.2 for the derivation of this equation) and output the most likely $u$. Note that due to the exponential complexity in $d$ of the algorithm, it is unfeasible to reach large values for this parameter. However, even at small $d$ values, the success rate for retrieving the correct spike has a very steep increase at around $\theta = 1$, as predicted by our analysis of the LR norm in theorem 2.

### 3.2 Computational distinguishability: LDLR analysis

We now compare the statistical threshold of section 3.1 with the *computational sample complexity exponent* that quantifies the sample complexity of detecting non-Gaussianity with an efficient algorithm that runs in a time that is polynomial in the input dimension. The algorithmic sample complexity can be analysed rigorously for a wide class of algorithms using the *low-degree method* [5–8, 37]. The low-degree method arose from the study of the sum-of-squares hierarchy [5] and rose to prominence when Hopkins & Steurer [7] demonstrated that the method can capture the Kesten–Stigum threshold for community detection in the stochastic block model [68–70]. In the case of hypothesis testing, the key quantity is the *low-degree likelihood ratio* (LDLR) [8, 37].

**Definition 3** (Low-degree likelihood ratio (LDLR)). *Let $D \geq 0$ be an integer. The low-degree likelihood ratio of degree $D$ is defined as*

$$L^{\leq D} := \mathcal{P}^{\leq D} L$$

(9)

*where $\mathcal{P}^{\leq D}$ projects $L$ onto the space of polynomials of degree up to $D$, parallel to the Hilbert space structure defined by the scalar product $\langle f, g \rangle_{L^2(\Omega, \mathbb{Q})} := \mathbb{E}_{x \sim \mathbb{Q}}[f(x)g(x)]$.*

The idea of this method is that among degree-$D$ polynomials, $L^{\leq D}$ captures optimally the difference between $\mathbb{P}$ and $\mathbb{Q}$, and this difference can be quantified by the norm $\left\| L^{\leq D} \right\| = \left\| L^{\leq D} \right\|_{L^2(\Omega_n, \mathbb{Q}_n)}$. Hence in analogy to the *second moment method* used in section 3.1, we can expect low-degree polynomials to be able to distinguish $\mathbb{P}$ from $\mathbb{Q}$ only when $\left\| L_n^{\leq D(n)} \right\| \underset{n}{\to} \infty$, where $D(n)$ is a monotone sequence diverging with $n$. Indeed, the following (informal) conjecture from Hopkins [8] states that this criterion is valid not only for polynomial tests, but for all polynomial-time algorithms:

**Conjecture 4.** *For two sequences of measures $\mathbb{Q}_N, \mathbb{P}_N$ indexed by $N$, suppose that (i) $\mathbb{Q}_N$ is a product measure; (ii) $\mathbb{P}_N$ is* sufficiently *symmetric with respect to permutations of its coordinates; and (iii) $\mathbb{P}_N$ is* robust *with respect to perturbations with small amount of random noise. If $\|L_N^{\leq D}\| = O(1)$ as $N \to \infty$ and for $D \geq (\log N)^{1+\varepsilon}$, for some $\varepsilon > 0$, then there is no polynomial-time algorithm that strongly distinguishes the distributions $\mathbb{Q}$ and $\mathbb{P}$.*

Even though this conjecture is still not proved in general, its empirical confirmation on many benchmark problems has made it an important tool to probe questions of computational complexity, see theorem 8 for a simple example.

We will now compute LDLR norm estimates for the spiked cumulant model, so that the application of conjecture 4 will help to understand the *computational sample complexity* of this model.

**Theorem 5** (LDLR for spiked cumulant model). *Suppose that $(u_i)_{i=1,\dots,d}$ are drawn i.i.d. from the symmetric Rademacher distribution and that the non-Gaussian distribution $p_g$ satisfies assumption 1. Let $0 < \varepsilon < 1$ and assume $D(n) \asymp \log^{1+\varepsilon}(n)$. Take $n, d \to \infty$, with the scaling $n \asymp d^\theta$ for $\theta > 0$. The following bounds hold:*

$$\left\| L_{n,d}^{\leq D(n)} \right\|^2 \geq \left( \frac{1}{\lfloor D(n)/4 \rfloor} \left( \frac{\beta^2 \kappa_4^g}{(1+\beta)^2} \right)^2 \frac{n}{d^2} \right)^{\lfloor D(n)/4 \rfloor}$$

(10)

$$\left\| L_{n,d}^{\leq D(n)} \right\|^2 \leq 1 + \sum_{m=1}^{D(n)} \left( \frac{\Lambda^2 \beta}{1+\beta} \right)^m m^{4m} \left( \frac{n}{d^2} \right)^{m/4}$$

(11)

*Taken together, eq. (10) and eq. (11) imply the presence of a critical regime for $\theta_c = 2$, and describe the behaviour of $\left\| L_{n,d}^{\leq D} \right\|$ for all $\theta \neq \theta_c$*

$$\lim_{n,d \to \infty} \left\| L_{n,d}^{\leq D(n)} \right\| = \begin{cases} 1 & 0 < \theta < 2 \\ +\infty & \theta > 2 \end{cases} \tag{12}$$

This theorem could be derived with different constants from lemma 26 and proposition 8 in Dudeja & Hsu [12]. Here we also provide a different, more direct argument. We sketch the proof of theorem 5 in appendix B.6.1 and give the complete proof in appendix B.6.2. We will discuss next the implications of the results presented in section 3.1 and section 3.2

### 3.3 Statistical-to-computational gaps in the spiked cumulant model

Put together, our results for the statistical and computational sample complexities of detecting non-Gaussianity in theorem 2 and theorem 5 suggest the existence of three different regimes in the spiked cumulant model as we vary the exponent $\theta$, with a statistical-to-computational gap: for $0 \leq \theta < 1$, the problem is *statistically impossible* in the sense that no algorithm is able to strongly distinguish $\mathbb{P}$ and $\mathbb{Q}$ with so few samples, since the LR norm is bounded. For $1 < \theta < 2$, the norm of the likelihood ratio with $g \sim \text{Rademacher}(1/2)$ diverges (even at $\theta = 1$ for some values of $\beta$), the problem could be statistically solvable (as validated by the results of exhaustive-search algorithms in fig. 1), but conjecture 4 suggests no polynomial-time algorithm is able to achieve distinguishability in this regime; this is the so-called *hard phase*. If $\theta > 2$, the problem is solvable in polynomial time by evaluating a polynomial function (fourth-order in each sample) and thresholding; this is the *easy phase*.

The spiked cumulant model leads thus to intrinsically harder classification problems than the spiked Wishart model, where the critical regime is at $\theta = 1$. The proof of theorem 5 reveals that this increased difficulty is a direct consequence of the whitening of the inputs in eq. (3). Without whitening, degree-2 polynomials would also give contributions to the LDLR (75) which would yield linear sample complexity. The difference in sample complexity of the spiked Wishart and spiked cumulant models mirrors the gap between the sample complexity of the best-known algorithms for matrix factorisation, which require linear sample complexity, and tensor PCA for rank-1 spiked tensors of order $k$ [25, 28, 30], where sophisticated spectral algorithms can match the computational lower bound of $d^{k/2}$ samples.

## 4 Learning from HOCs with neural networks and random features

The analysis of the (low-degree) likelihood ratio has given us a detailed picture of the statistical and computational complexity of extracting information from the covariance or the higher-order cumulants of data. We will now use these fundamental limits to benchmark the sample complexity of two-layer neural networks (2LNN) trained with stochastic gradient descent on a binary discrimination task, where inputs in one class are drawn from the normal distribution $\mathcal{N}(0, \mathbb{1}_d)$, while inputs in the other class are drawn from the spiked Wishart or the spiked cumulant model. In addition, we will also benchmark random feature methods (RF) [71–73] as a finite-dimensional approximation of kernel methods [71–73].

In a nutshell, the idea behind our experiments is to first *validate* both 2LNN and RF on the simpler spiked Wishart task and then to apply both methods to the spiked cumulant model, where inputs are generated in a way that mirrors the spiked Wishart: comparing eq. (1) with eqs. (2) and (3), we see that the only differences are the whitening, and the latent distribution $p_g$. In our experiments, we choose the latent variables to be standard Gaussian for spiked Wishart, and Rademacher for the spiked cumulant – hence the latent variables have matching first and second moments. However, we will see that the spiked cumulant model exhibits a large gap in the sample complexity required for neural networks or random features to learn the problem. We relegate details on the experimental setups such as hyper-parameters to appendix A.

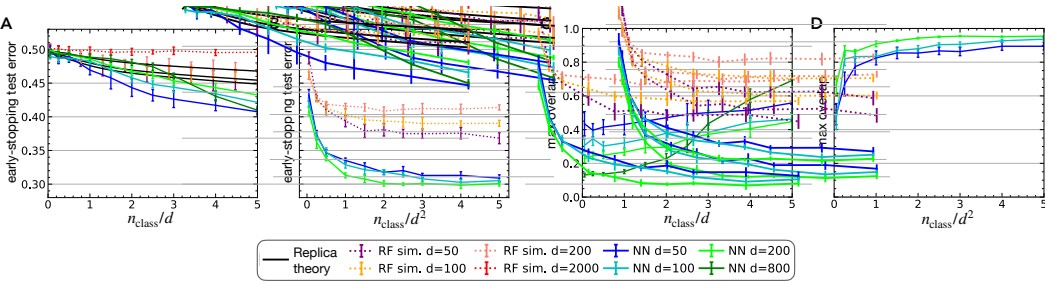

Figure 2: **Learning the spiked Wishart task with neural networks and random features. (A,B)** Test accuracy of random features (RF) and early-stopping test accuracy of two-layer ReLU neural networks (NN) on the spiked Wishart task, eq. (1), with linear and quadratic sample complexity ($n_{\text{class}} \asymp d, d^2$, respectively, where $d$ is the input dimension). Predictions for the performance of random features obtained using replicas are shown in black. **(C,D)** Maximum normalised overlaps of the networks' first-layer weights with the spike $u$, eq. (1). *Parameters*: $\beta = 5$. Neural nets and random features have $m = 5d$ hidden neurons. Full experimental details in appendix A.

## 4.1 Spiked Wishart model

We trained **two-layer ReLU neural networks** $\phi_\theta(x) = v^\top \max\left(0, w^T x\right)$ with $m = 5d$ hidden neurons on the spiked Wishart classification task. We show the early-stopping test accuracy of the networks in the linear and quadratic regimes with $n_{\text{class}} \asymp d, d^2$ samples per class in fig. 2A and B, resp. Neural networks are able to solve this task, in the sense that their classification error *per sample* is below 0.5, implying strong distinguishability of the whole data matrix. Indeed, some of the hidden neurons converge to the spike $u$ of the covariance matrix, as can be seen from the maximum value of the normalised overlaps $\max_k w_k^\top u / \sqrt{\|w_k\| \|u\|}$ in fig. 2C and D, where $w_k$ is the weight vector of the $k$th hidden neuron. In the linear regime (C), there is an increasingly clear transition from a random overlap for small data sets to a large overlap at large data sets as we increase the input dimension $d$; in the quadratic regime (D), the neural network recovers the spike almost perfectly.

The **relatively large overlap between hidden neurons and spike at small sample complexities** (fig. 2C and D) is due to the fact that we plot the maximum overlaps over a relative large number of hidden neurons $m = 5d$; hence even at initialisation, a few neurons will have large overlaps. We verified that ensuring an initial overlap of only $1/\sqrt{d}$ by explicit orthogonalisation did not change our results on distinguishability, see fig. 5. A possible explanation is that the dynamics of the wide network is dominated by the majority of neurons, which do not have a macroscopic overlap.

Meanwhile, we found that **the performance of random features** tends to random guessing at linear sample complexity, where we let the input dimension $d \to \infty$ with $m/d$ and $n_{\text{class}}/d$ fixed, while at quadratic sample complexity, random features learn the task, although they perform worse than neural networks. The failure of RF in the linear regime makes sense in light of recent results that suggest that random features in this scaling regime are limited to learning a linear approximation of the target function [74–77], while the LDLR analysis appendix B.2.3 shows that the target function, i.e. the low-degree likelihood ratio, is a quadratic polynomial. However, these results are, to the best of our knowledge, restricted to the case of Gaussian isotropic inputs.

To ensure that the performance of random features does indeed tend to random guessing, we performed a **replica analysis** following Loureiro *et al.* [78] for mixture classification tasks together with the Gaussian equivalence theorem [79–83] (black lines in fig. 2A, details in appendix C). Replica theory perfectly predicts the performance of RF we obtain in numerical experiments (red-ish dots) for various values of $d$ at linear sample complexity. We thus find a clear separation in the sample complexity required by random features ($n_{\text{class}} \gtrsim d^2$) and neural networks ($n_{\text{class}} \gtrsim d$) to learn the spiked Wishart task. The replica analysis can be extended to the polynomial regime by a simple rescaling of the free energy [84] on several data models, like the vanilla teacher–student setup [85], the Hidden manifold model [79], and the vanilla Gaussian mixture classification (see fig. 8). However, we found that for the spiked Wishart model, the Gaussian equivalence theorem which we need to deal with the random feature distribution *fails* at quadratic sample complexity. This might be due to the fact that in this case, the spike induces a dominant direction in the covariance of the random features, and this

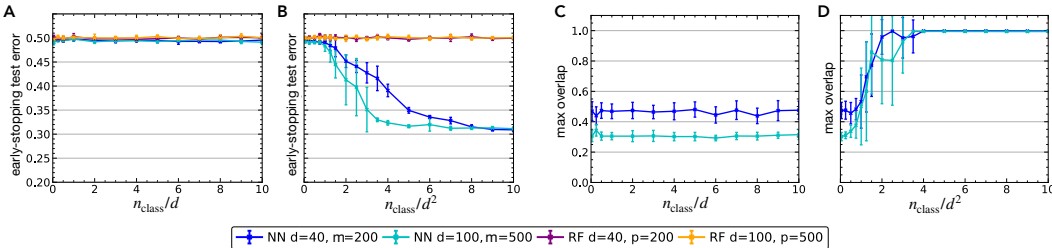

Figure 3: **Learning the spiked cumulant task with neural networks and random features. (A, B)** Test accuracy of random features (RF) and early-stopping test accuracy of two-layer ReLU neural networks (NN) on the spiked cumulant task eq. (3) with linear and quadratic sample complexity ($n_{\text{class}} \asymp d, d^2$, respectively, where $d$ is the input dimension). **(C, D)** Maximum normalised overlaps of the networks' first-layer weights with the spike $u$, (3). *Parameters*: $\beta = 10$. Neural nets and random features have $m = 5d$ hidden neurons, same optimisation as in fig. 2. Full experimental details in appendix A.

direction is also key to solving the task, which can lead to a breakdown of Gaussian equivalence [83]. A similar breakdown of the Gaussian equivalence theorem at quadratic sample complexity has been demonstrated recently in teacher–student setups by Cui *et al.* [86] and Camilli *et al.* [87].

### 4.2 Spiked cumulant

Having thus validated the performance of 2LNN and RF on the simpler spiked Wishart model, we turn to the spiked cumulant model and to the question: can neural networks learn higher-order correlations, efficiently? The LDLR analysis predicts that a polynomial-time algorithm requires at least a quadratic number of samples to detect non-Gaussianity, and hence to solve the classification task, and we found indeed that neural networks require at least quadratic sample complexity to solve the task, fig. 3A and B. The high values of the maximum overlap between hidden neurons and cumulant spike in the linear regime (compared to $d^{-1/2}$) are again a consequence of choosing the maximum overlap among $m = 5d$ hidden neurons. Random features cannot solve this task even at quadratic sample complexity, since they are limited to a quadratic approximation of the target function [75–77, 88], but we know from the LDLR analysis that the target function is a fourth-order polynomial. We thus find an even larger separation in the minimal number of samples required for random features and neural networks to solve tasks that depend on directions which are encoded exclusively in the higher-order cumulants of the inputs.

### 4.3 Phase transitions and neural network performance in a simple model for images

We finally show another example of a separation between the performance of random features and neural networks in the feature-learning regime on a toy model for images that was introduced recently by Ingrosso & Goldt [13], the non-linear Gaussian process (NLGP). The idea is to generate inputs that are (i) translation-invariant and that (ii) have sharp edges, both of which are hallmarks of natural images [89]. We first sample a vector $z \in \mathbb{R}^d$ from a normal distribution with zero mean and covariance $C_{ij} = \mathbb{E} z_i z_j = \exp(-|i - j|/\xi)$ to ensure translation-invariance of the inputs, with length scale $\xi > 0$. We then introduce edges, i.e. sharp changes in luminosity, by passing $z$ through a saturating non-linearity like the error function, $x_i = \text{erf}(g z_i)/Z(g)$, where $Z(g)$ is a normalisation factor that ensures that the pixel-wise variance $\mathbb{E} x_i^2 = 1$ for all values of the gain $g > 0$. The classification task is to discriminate these "images" from Gaussian inputs with the same mean and covariance, as illustrated in two dimensions in fig. 4A. This task is different from the spiked cumulant model in that the cumulant of the NLGP is not low-rank, so there are many directions that carry a signal about the non-Gaussianity of the inputs.

We trained wide two-layer neural networks on this task and interpolated between the feature-learning and the "lazy" regimes using the $\alpha$-renormalisation trick of Chizat *et al.* [90]. As we increase $\alpha$, the networks go from feature-learners ($\alpha = 1$) to an effective random feature model and require an increasing amount of data to solve the task, fig. 4B. There appears to be a sharp transition from random guessing to non-trivial performance as we increase the number of training samples for all values of $\alpha$.

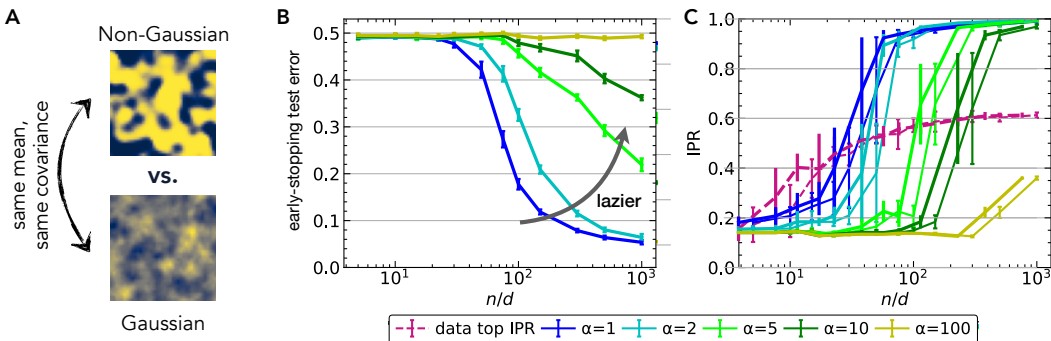

Figure 4: **A phase transition in the fourth-order cumulant precedes learning from the fourth cumulant.** **(A)** We train neural networks to discriminate inputs sampled from a simple non-Gaussian model for images introduced by Ingrosso & Goldt [13] (top) from Gaussians with the same mean and covariance (bottom). **(B)** Test error of two-layer neural networks interpolating between the fully-trained ($\alpha = 1$) and lazy regimes (large $\alpha$) – see section 4.3. **(C)** The localisation of the leading CP-factor of the non-Gaussian inputs (dashed purple line) and the first-layer weights of the trained networks, as measured by the inverse participation ratio (IPR), eq. (13). Large IPR denotes a more localised vector $w$. *Parameters*: $g = 3, \xi = 1, d = 20, m = 100$. Full details in appendix A.

This transition is preceded by a transition in the behaviour of the higher-order cumulant that was reported by Ingrosso & Goldt [13]. They showed numerically that the CP-factors of the empirical fourth-order cumulant $T$, defined as the vectors $\hat{u} \in \mathbb{R}^d$ that give the best rank-$r$ approximation $\hat{T} = \sum_{k=1}^{r} \gamma_k \hat{u}_k^{\otimes 4}$ of $T$ [91], localise in space if the data set from which the empirical cumulant is calculated is large enough. Quantifying the localisation of a weight vector $w$ using the *inverse participation ratio*

$$\text{IPR}(w) = \frac{\sum_{i=1}^{d} w_i^4}{\left(\sum_{i=1}^{d} w_i^2\right)^2}, \tag{13}$$

we confirm that the leading CP-factors of the fourth-order cumulant localise (purple dashed line in fig. 4C). The localisation of the CP-factors occurs with slightly less samples than the best-performing neural network requires to learn ($\alpha = 1$). The weights of the neural networks also localise at a sample complexity that is slightly below the sample complexity for solving the task. The laziest network ($\alpha = 100$), i.e. the one where the first-layer weights move least and which is hence closest to random features, does not learn the task even with a training set containing $n = 10^3 d$ samples when $d = 20$, indicating again a large advantage of feature-learners over methods with fixed feature maps, such as random features.

## 5  Concluding perspectives

Neural networks crucially rely on the higher-order correlations of their inputs to extract statistical patterns that help them solve their tasks. Here, we have studied the difficulty of learning from higher-order correlations in the spiked cumulant model, where the first non-trivial information in the data set is carried by the input cumulants of order 4 and higher. Our LR analysis of the corresponding hypothesis test confirmed that data sampled from the spiked cumulant model could be statistically distinguishable (in the sense that it passes the *second moment method for distinguishability*) from isotropic Gaussian inputs at linear sample complexity, while the number of samples required to strongly distinguish the two distributions in polynomial time scales as $n \gtrsim d^2$ for the class of algorithms covered by the low-degree conjecture [5–8], suggesting the existence of a large statistical-to-computational gap in this problem. Our experiments with neural networks show that they learn from HOCs efficiently in the sense that they match the sample complexities predicted by the analysis of the hypothesis test, which is in stark contrast to random features, which require a lot more data. In the future, a key challenge will be extend this framework to null hypotheses that go beyond isotropic Gaussian distributions. It will be intriguing to analyse the *dynamics* of neural networks on spiked cumulant models or the non-linear Gaussian process to understand how neural networks extract information from the higher-order cumulants of realistic data sets efficiently [92].

## Acknowledgements

We thank Zhou Fan, Yue Lu, Antoine Maillard, Alessandro Pacco, Subhabrata Sen, Gabriele Sicuro, and Ludovic Stephan for stimulating discussions on various aspects of this work. SG acknowledges co-funding from Next Generation EU, in the context of the National Recovery and Resilience Plan, Investment PE1 – Project FAIR "Future Artificial Intelligence Research", and from the European Union - NextGenerationEU, in the framework of the PRIN Project SELFMADE (code 2022E3WYTY – CUP G53D23000780001).

## Contributions

ES performed the numerical experiments with neural networks and random features. LB performed the (low-degree) likelihood analysis. FG performed the replica analysis of random features. SG designed research and advised ES and LB. All authors contributed to writing the paper.

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

# Appendix

## A    Experimental details

### A.1    Figures 2 and 3

For the spiked Wishart and spiked cumulant tasks, we trained two-layer ReLU neural networks $\phi_\theta(x) = v^\top \max\left(0, w^T x\right)$. The number of hidden neurons $m = 5d$, where $d$ is the input dimension. We train the networks using SGD with a learning rate of 0.002 and a weight-decay of 0.002 for 50 epochs for the spiked Wishart task and for 200 epochs for the spiked cumulant task. The plots show the early-stopping test errors. The results are plotted as averages over 10 random seeds, showing the standard deviation by the errorbars.

The random features (RF) models [72, 73] have also a width of $5d$. The ridge regression is performed using scikit-learn [93] with a regularisation of 0.1.

For the spiked datasets, the spikes are from the Rademacher distribution, using a signal-to-noise ratio of 5.0 for the spiked Wishart and 10.0 for the spiked cumulant datasets. For the overlaps between spikes and features overlaps, we plot the highest overlap amongst the incoming weights of the hidden neurons with the spike, calculated as the normalised dot product.

**Starting from small initial overlap**    Since the neural networks have a large number of neurons ($m = 5d$), some of them will have a relatively large overlap with the spike at initialisation, as can be seen in fig. 3. To check that this relatively large overlap did not affect our results, we repeated the same set of experiments while enforcing an overlap of all hidden neuron weights with the spike of $1/\sqrt{d}$ by explicit orthogonalisation, as discussed in section 4.1. We show the results in fig. 5: while the maximum overlaps do indeed decrease for small sample complexities, the qualitative behaviour is unchanged.

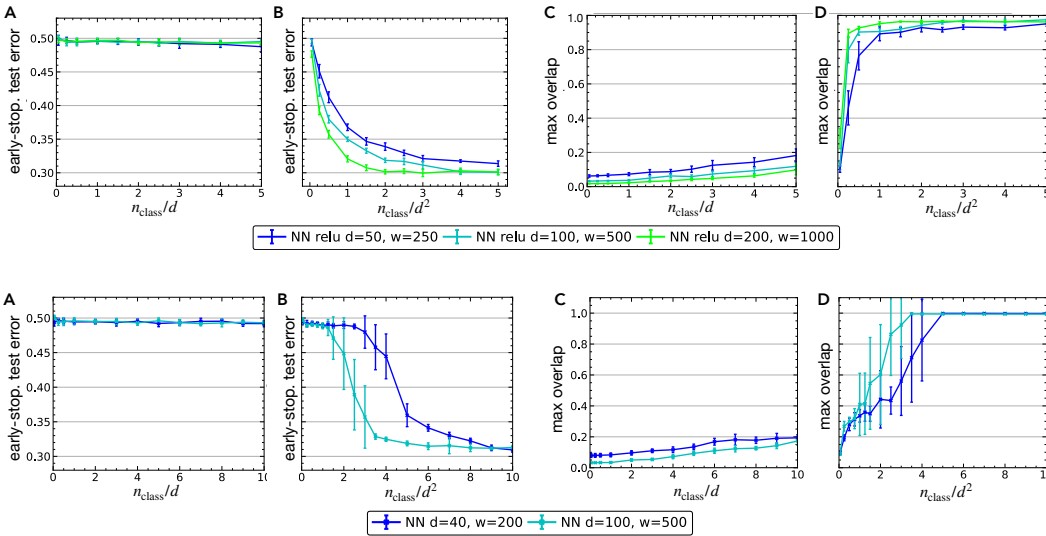

Figure 5: **Learning the spiked Wishart and spiked cumulant task, starting from small initial overlaps.** We repeat the neural network experiments on the spiked Wishart (top) and spiked cumulant (bottom) task, see figs. 2 and 3, while enforcing that all hidden neurons have an overlap of exactly $1/\sqrt{d}$ with the spikes, by simple explicit orthogonalisation. While the maximum overlaps do indeed decrease for small sample complexities, the qualitative behaviour is unchanged. All hyper-parameters as in figs. 2 and 3, respectively.

## A.2 Figure 4

For the NLGP–GP task, we use the $\alpha$-scaling trick of Chizat *et al.* [90] to interpolate between feature- and lazy-learning. We define the network function as:

$$\phi_{\text{NN}}(x; v, W, v_0, W_0) = \frac{\alpha_{\text{NTK}}}{K} \left[ \sum_j^m v_j \sigma \left( \sum_i^d w_i x_i \right) - \sum_j^m v_{0,j} \sigma \left( \sum_i^d w_{0,i} x_i \right) \right] \quad (14)$$

where $v_0, w_0$ are kept fixed at their initial values. The mean-squared loss is also rescaled by $(1/\alpha_{\text{NTK}}^2)$. Changing $\alpha_{\text{NTK}}$ from low to high allows to interpolate between the feature- and the lazy-learning limits as the first-layer weights will move away less from their initial values.

For fig. 4, the network has a width of 100 and the optimisation is run by SGD for 200 epochs with a weight-decay of $5 \cdot 10^{-6}$ and learning rate of $0.5$. The one-dimensional data vectors have a length of 20; the correlation length is 1 and the signal-to-noise ratio is set to 3. The error shown is early-stopping test error. The localisation of the neural networks' features and the data's fourth moments is shown by the IPR measure. (We used here a lower length for the data vectors so that the calculations for fourth-order cumulants do not have high memory requirements.) For the neural networks, the highest IPR is shown amongst the incoming features of the hidden neurons at the final state of the training. For the data, the highest IPR is the highest amongst the CP-factors of the fourth cumulants of the nlgp-class using a rank-1 PARAFAC decomposition from the tensorly package [94] .

# B  Mathematical details on the hypothesis testing problems

In this section, we provide more technical details on Hermite polynomials, LR and LDLR, and present the complete proofs of all theorems stated in the main.

## B.1  Notation

We use the convention $0 \in \mathbb{N}$. $n \in \mathbb{N}$ denotes the number of samples, $d$ the dimensionality of each data point $x$. The letters $k, m$ usually denote free natural parameters. Let $[n] := \{1, \ldots, n\}$, $\mu \in [n]$ is an index that ranges on the samples and $i \in [d]$ usually ranges on the data dimension. Underlined letters $\underline{x}, \mathbb{P}, \mathbb{Q}$ are used for objects that are $n \times d$ dimensional. In proofs the letter $C$ denotes numerical constants whose precise value may change from line to line, and any dependency is denoted as a subscript. $m!$ denotes factorial and $m!!$ denotes the double factorial (product of all the numbers $k \leq m$ with the same parity as $m$).

**Multi-index notation**  We will need multi-index notation to deal with $d$-dimensional functions and polynomials. Bold greek letters are used to denote multi-indices and their components, $\boldsymbol{\alpha} = (\boldsymbol{\alpha}_1, \ldots, \boldsymbol{\alpha}_d) \in \mathbb{N}^d$. The following conventions are adopted:

$$|\boldsymbol{\alpha}| := \sum_{i=1}^d \boldsymbol{\alpha}_i, \quad \boldsymbol{\alpha}! := \prod_{i=1}^d \boldsymbol{\alpha}_i!, \quad x^{\boldsymbol{\alpha}} := \prod_{i=1}^d x_i^{\boldsymbol{\alpha}_i}$$

$$\partial_{\boldsymbol{\alpha}} f(x_1, \ldots, x_d) = \frac{\partial}{\partial x^{\boldsymbol{\alpha}}} f := \left( \frac{\partial}{\partial x_1^{\boldsymbol{\alpha}_1}} \cdots \frac{\partial}{\partial x_d^{\boldsymbol{\alpha}_d}} \right) f \quad (15)$$

Since we will have $n$ samples of $d$-dimensional variables, we will need to consider polynomials and functions in $nd$ variables. To deal with all these variables we introduce multi-multi-indices (denoted by underlined bold Greek letters $\underline{\boldsymbol{\alpha}}, \underline{\boldsymbol{\beta}}, \ldots$). They are $n \times d$ matrices with entries in $\mathbb{N}$ (i.e. elements of $\mathbb{N}^{n \times d}$)

$$\underline{\boldsymbol{\alpha}} := (\boldsymbol{\alpha}_i^{\mu}) = \begin{pmatrix} \boldsymbol{\alpha}_1^1 & \cdots & \boldsymbol{\alpha}_d^1 \\ \vdots & \ddots & \vdots \\ \boldsymbol{\alpha}_1^n & \cdots & \boldsymbol{\alpha}_d^n \end{pmatrix} \quad (16)$$

We denote by $\boldsymbol{\alpha}^{\mu}$ the rows of $\underline{\boldsymbol{\alpha}}$, that are $d$-dimensional multi-indices.

All the notations (15) generalize to multi-multi-indices in the following way:

$$|\underline{\boldsymbol{\alpha}}| := \sum_{\mu=1}^{n} |\boldsymbol{\alpha}^{\mu}| = \sum_{\mu=1}^{n} \sum_{i=1}^{d} \boldsymbol{\alpha}_i^{\mu}, \quad \underline{\boldsymbol{\alpha}}! := \prod_{\mu} \boldsymbol{\alpha}^{\mu}! = \prod_{\mu} \prod_i \boldsymbol{\alpha}_i^{\mu}!,$$

$$\underline{x}^{\underline{\boldsymbol{\alpha}}} := \prod_{\mu=1}^{n} (x^{\mu})^{\boldsymbol{\alpha}^{\mu}} = \prod_{\mu=1}^{n} \prod_{i=1}^{d} (x_i^{\mu})^{\boldsymbol{\alpha}_i^{\mu}} \tag{17}$$

$$\partial_{\underline{\boldsymbol{\alpha}}} f(\underline{x}) := \left( \frac{\partial}{\partial (x^1)^{\boldsymbol{\alpha}^1}} \cdots \frac{\partial}{\partial (x^n)^{\boldsymbol{\alpha}^n}} \right) f(\underline{x})$$

## B.2 More on LR and LDLR

### B.2.1 Statistical and computational distinguishability

From a statistical point of view, distinguishing two sequences of probability measures $\underline{\mathbb{P}} = (\mathbb{P}_n)_{n \in \mathbb{N}}$ and $\underline{\mathbb{Q}} = (\mathbb{Q}_n)_{n \in \mathbb{N}}$ defined on a sequence of measure spaces $(\Omega_n, \mathcal{F}_n)$ means finding a sequence of tests $f_n : \Omega_n \to \{0, 1\}$, which are measurable functions that indicate whether a given set of inputs was sampled from $\mathbb{P}$ or $\mathbb{Q}$. We will say that the two measures are **statistically distinguishable** in the strong sense if there exists a statistical test $f$ for which

$$\mathbb{Q}_n \left( f_n(x) = 0 \right) \underset{n \to \infty}{\longrightarrow} 1 \quad \text{and} \quad \mathbb{P}_n (f_n(x) = 1) \underset{n \to \infty}{\longrightarrow} 1. \tag{18}$$

The strong version of statistical distinguishability requires that the probability of success of the statistical test must tend to 1 as $n \to \infty$, whereas **weak distinguishability** just requires it to be asymptotically greater than $1/2 + \varepsilon$, for any $\varepsilon > 0$. We will call the minimal number of samples required to achieve strong statistical distinguishability the **statistical sample complexity** of the problem. We can obtain a computationally bounded analogue of this definition by restricting the complexity of the statistical test $f_n$ to functions that are computable in a time that is polynomial in the input dimension $d$. The **computational statistical complexity** is then the minimal number of samples required to achieve **computational distinguishability**.

### B.2.2 Necessary conditions for distinguishability

A necessary condition for strong distinguishability is based on the *likelihood ratio* (LR) of probability measures, which is defined as

$$L_n(x) := \frac{d\mathbb{P}_n}{d\mathbb{Q}_n}(x) \tag{19}$$

The likelihood ratio provides a necessary condition for strong distinguishability of $\mathbb{P}$ and $\mathbb{Q}$ via the so-called second moment method:

**Proposition 6** (Second Moment Method for Distinguishability). *Suppose that $\mathbb{P}_n$ is absolutely continuous with respect to $\mathbb{Q}_n$, and let $L_n$ be the corresponding LR. A necessary condition for strong distinguishability of $\underline{\mathbb{P}}$ from $\underline{\mathbb{Q}}$ is*

$$\|L_n\|^2 := \underset{x \sim \mathbb{Q}_n}{\mathbb{E}} [L_n(x)^2] \underset{n \to \infty}{\longrightarrow} +\infty. \tag{20}$$

*where $\|\cdot\|$ is the norm with respect to the Hilbert space*

$$L^2(\Omega_n, \mathbb{Q}_n) = \left\{ f : \Omega_n \to \mathbb{R} \mid \mathbb{E}_{\mathbb{Q}_n}[f^2(x)] < \infty \right\}. \tag{21}$$

*Proof.* The proof is immediate: if $(\|L_n\|)_n$ were bounded, by Cauchy-Schwartz $\mathbb{Q}(A_n) \to 0$ would imply $\mathbb{P}(A_n) \to 0$. But by the definition of strong asymptotic distinguishability there exists a sequence of events $(A_n)_n$ such that $\mathbb{P}(A_n) \to 1$ and $\mathbb{Q}(A_n) \to 0$; hence $(\|L_n\|)_n$ must diverge. $\square$

The second moment method has been used to derive statistical thresholds for various high-dimensional inference problems [22, 27, 37, 95]. Note that proposition 6 only states a necessary condition; while it is possible to construct counterexamples, they are usually based on rather artificial constructions (like the following example 1 from [37]), and the second moment method is considered a good proxy for strong statistical distinguishability.

**Example 7.** *Let $\mathbb{P}$ and $\mathbb{Q}$ be two strongly distinguishable sequences of probability measures on $(\mathcal{Y}_n, \mathcal{F}_n)_n$. Define $\mathbb{P}'_n$ on $\mathcal{F}_n$ as a measure that with probability $1/2$ follows $\mathbb{P}_n$ and with probability $1/2$ follows $\mathbb{Q}_n$. Letting $L'_n$ be the LR of $\mathbb{P}'_n$ with respect to $\mathbb{Q}_n$. We have that $||L'_n|| \to \infty$, but $\underline{\mathbb{P}}'$ and $\mathbb{Q}$ are not strongly distinguishable.*

*Proof.* The density of $\mathbb{P}'_n$ is:

$$p'_n(y) = \frac{p_n(y) + q_n(y)}{2} \tag{22}$$

hence

$$L'_n = \frac{1}{2}\left(1 + L_n\right) \tag{23}$$

which clearly has the same asymptotic behaviour as $L_n$.

On the other hand, it cannot be that $\underline{\mathbb{P}}'$ and $\underline{\mathbb{Q}}'$ are strongly distinguishable since for any event $A$, $\mathbb{P}'_n(A) \geq \frac{1}{2}\mathbb{Q}_n(A)$. $\qquad\square$

### B.2.3 LDLR analysis for spiked Wishart model

In this subsection we show the application of LDLR method to the spiked Wishart model. We obtain the correct BBP threshold for the signal-to-noise ratio even when we restrict ourselves to the class of polynomials of constant degree:

**Theorem 8** (LDLR for spiked Wishart model). *Suppose the prior on $u$ belongs to the following cases:*

- *$(u_i)_{i=1,\ldots,d}$ are i.i.d. and symmetric Rademacher random variables*

- *$(u_i)_{i=1,\ldots,d}$ are i.i.d. and $u_i \sim \mathcal{N}(0,1)$*

*Let $D \in \mathbb{N}$ and $d, n \to \infty$, with fixed ratio $\gamma := d/n$, then*

$$\lim_{d,n\to\infty} \|L^{\leq D}\| = \sum_{k=0}^{D} \frac{(2k-1)!!}{(2k)!!} \frac{\beta^{2k}}{\gamma^k}, \tag{24}$$

*which, as $D$ increases, stays bounded for $\beta < \beta_c := \sqrt{\gamma} = \sqrt{d/n}$ and diverges for $\beta > \beta_c$.*

The distinguishability threshold that we have recovered here is of course the famous BBP phase transition in the spiked Wishart model [9]: if $\beta < \beta_c = \sqrt{d/n}$, the low-degree likelihood ratio stays bounded and indeed a set of inputs drawn from $1$ is statistically indistinguishable from a set of inputs drawn from $\mathcal{N}(0, \mathbb{1})$. For $\beta > \beta_c$ instead, there is a phase transition for the largest eigenvalue of the empirical covariance of the inputs in the second class, which can be used to differentiate the two classes, and the LDLR diverges.

As mentioned in the main text, a thorough LDLR analysis of a more general model that encompasses the spiked Wishart model was performed by Bandeira *et al.* [96]. Their theorem 3.2 is a more general version of our theorem 8, that generalizes it in two directions: it works for a class of subgaussian priors on $u$ that satisfy some concentration property, and also allows for negative SNR (the requirement is $\beta > -1$). For completeness, here we show that this result can be obtained as a straightforward application of a Gaussian additive model.

*Proof.* We note that the problem belongs the class of *Additive Gaussian Noise models* which is studied in depth by Kunisky *et al.* [37]. In those models the two hypotheses have to be expressed as an additive perturbation of white noise:

- $\mathbb{P}_n$: $\underline{x}_n = y_n + z_n$,
- $\mathbb{Q}_n$: $\underline{x}_n = z_n$,

The spiked Wishart model that we consider belongs to this class. It can be seen by defining $\mathbb{R}^{nd} \ni y_n = \left(\sqrt{\frac{\beta}{d}}g^1 u, \ldots, \sqrt{\frac{\beta}{d}}g^n u\right)^\top$. So we can apply theorem 2.6 from Kunisky *et al.* [37], that

computes the norm of the LDLR by using two independent replicas of the variable $y$. Denoting the two replicas by $\hat{y}$ and $\tilde{y}$, we get

$$
\begin{aligned}
||L_{\hat{n}}^{\leq D}||^2 &= \mathbb{E}\left[\sum_{m=0}^{D} \frac{1}{m!}(\hat{y}\cdot\tilde{y})^m\right] \\
&= \mathbb{E}\left[\sum_{m=0}^{D} \frac{\beta^m}{m!d^m}\left(\sum_{\mu=1}^{n}\hat{g}^\mu\tilde{g}^\mu\hat{u}\cdot\tilde{u}\right)^m\right] \\
&= \sum_{m=0}^{D} \frac{\beta^m}{m!d^m}\mathbb{E}\left[(\hat{u}\cdot\tilde{u})^m\left(\sum_{\mu=1}^{n}\hat{g}^\mu\tilde{g}^\mu\right)^m\right] \\
&= \sum_{m=0}^{D} \frac{\beta^m}{m!d^m}\mathbb{E}\left[\left(\sum_{\mu=1}^{n}\hat{g}^\mu\tilde{g}^\mu\right)^m\right]\mathbb{E}\left[(\hat{u}\cdot\tilde{u})^m\right]
\end{aligned}
\tag{25}
$$

Note now that $\sum_{\mu=1}^{n}\hat{g}^\mu\tilde{g}^\mu$ has the same distribution as $-\sum_{\mu=1}^{n}\hat{g}^\mu\tilde{g}^\mu$, so its distribution is even and all the odd moments are 0. This means that we can reduce to the case $m=2k$, so we need to study:

$$
||L_{\hat{n}}^{\leq D}||^2 = \sum_{k=0}^{\lfloor D/2\rfloor} \frac{\beta^{2k}}{(2k)!d^{2k}}\underbrace{\mathbb{E}\left[\left(\sum_{\mu=1}^{n}\hat{g}^\mu\tilde{g}^\mu\right)^{2k}\right]}_{T_1}\underbrace{\mathbb{E}\left[(\hat{u}\cdot\tilde{u})^{2k}\right]}_{T_2}
\tag{26}
$$

Let us consider the term $T_1$. Call $Y^\mu := \hat{g}^\mu\tilde{g}^\mu$. The distribution of each of the $Y^\mu$ is not Gaussian, but we have that $\mathbb{E}[Y^\mu]=0$ and $\mathrm{Var}(Y^\mu)=1$. So by the central limit theorem $S_n := \frac{1}{\sqrt{n}}\sum_{\mu=1}^{n}Y^\mu$ converges to a standard normal in distribution, as $n\to\infty$. Note that the cumulants of $S_n$ can be computed thanks to linearity and additivity of cumulants and are $\kappa_{2k}^{(S_n)} = n^{1-k}\kappa_{2k}^Y$. So they all go to 0 except from the variance. Since the moments can be written as a function of the cumulants up to that order, it follows that $\lim_n \mathbb{E}[S_n^{2k}]$ will be the $2k$-th moment of the standard normal distribution, which means that we have the following:

$$
\lim_{n\to+\infty}\mathbb{E}\left[\left(\frac{1}{\sqrt{n}}\sum_{\mu=1}^{n}\hat{g}^\mu\tilde{g}^\mu\right)^{2k}\right] = (2k-1)!!
\tag{27}
$$

We turn now to T2 and do the same reasoning. Define $v_i := \hat{u}_i\tilde{u}_i$, both in Rademacher and in Gaussian prior case, we have that the $(v_i)_{i=1,\dots,d}$ is an independent family of random variables that have 0 mean and variance equal to 1. So we can again apply the central limit theorem to get that:

$$
\lim_{d\to+\infty}\mathbb{E}\left[\left(\frac{\hat{u}\cdot\tilde{u}}{\sqrt{d}}\right)^{2k}\right] = (2k-1)!!
\tag{28}
$$

Taking the limit for $n,d\to+\infty$ with the constraint $\gamma = d/n$, we have that:

$$
\begin{aligned}
\lim_{n,d\to\infty}||L_{\hat{n}}^{\leq D}||^2 &= \lim_{n,d\to\infty}\sum_{k=0}^{\lfloor D/2\rfloor}\frac{\beta^{2k}n^k}{(2k)!d^k}\mathbb{E}\left[\left(\frac{1}{\sqrt{n}}\sum_{\mu=1}^{n}\hat{g}^\mu\tilde{g}^\mu\right)^{2k}\right]\mathbb{E}\left[\left(\frac{\hat{u}\cdot\tilde{u}}{\sqrt{d}}\right)^{2k}\right] \\
&= \sum_{k=0}^{D}\frac{\left(\beta^k(2k-1)!!\right)^2}{(2k)!\gamma^k} = \sum_{k=0}^{D}\frac{(2k-1)!!}{(2k)!!}\frac{\beta^{2k}}{\gamma^k}
\end{aligned}
\tag{29}
$$

which is what we wanted to prove.

As a final note we remark that the basic ideas of the arguments in [96] coincide with what exposed above. However, the increased generality of the statement requires the use of the abstract theory of Umbral calculus to generalize the notion of Hermite polynomials to negative SNR cases, as well as more technical work to achieve the bounds on the LDLR projections. $\qquad\square$

### B.3 Hermite Polynomials

We recall here the definitions and key properties of the Hermite polynomials.

**Definition 9.** *The Hermite polynomial of degree $m$ is defined as*

$$h_m(x) := (-1)^m e^{\frac{x^2}{2}} \frac{d^m}{dx^m} \left( e^{-\frac{x^2}{2}} \right) \tag{30}$$

Here is a list of the first 5 Hermite polynomials:

$$\begin{aligned}
h_0(x) &= 1 \\
h_1(x) &= x \\
h_2(x) &= x^2 - 1 \\
h_3(x) &= x^3 - 3x \\
h_4(x) &= x^4 - 6x^2 + 3
\end{aligned} \tag{31}$$

The Hermite polynomials enjoy the following properties that we will use in the subsequent proofs (for details see section 5.4 in McCullagh [97], Szegő [98] and Abramowitz & Stegun [99]):

- they are orthogonal with respect to the $L^2$ product weighted with the density of the Normal distribution:

$$\frac{1}{\sqrt{2\pi}} \int_{-\infty}^{\infty} h_n(x) h_m(x) e^{-\frac{x^2}{2}} \, dx = n! \delta_{m,n}; \tag{32}$$

- $h_m$ is a monic polynomial of degree $m$, hence $(h_m)_{m \in \{1,\dots,N\}}$ generates the space of polynomials of degree $\leq N$;

- the previous two properties imply that the family of Hermite polynomials is an orthogonal basis for the Hilbert space $L^2(\mathbb{R}, \mathbb{Q})$ where $\mathbb{Q}$ is the normal distribution;

- they enjoy the following recurring relationship

$$h_{m+1}(x) = x h_m(x) - h'_m(x), \tag{33}$$

which can also be expressed as a relationship between coefficients as follows. If $h_m(x) = \sum_{k=0}^{m} a_{m,k} x^k$, then

$$a_{m+1,k} = \begin{cases} -a_{m,1} & k = 0, \\ a_{m,k-1} - (k+1) a_{m,k+1} & k > 0. \end{cases} \tag{34}$$

They also satisfy identities of binomial type, like:

$$h_m(x+y) = \sum_{k=0}^{m} \binom{m}{k} x^{m-k} h_k(y) \tag{35}$$

$$h_m(\gamma x) = \sum_{j=0}^{\lfloor m/2 \rfloor} \gamma^{m-2j} (\gamma^2 - 1)^j \binom{m}{2j} (2j-1)!! \, h_{m-2j}(x) \tag{36}$$

**Multivariate case** In the multivariate $m$-dimensional case we can consider Hermite tensors $(H_\alpha)_{\alpha \in \mathbb{N}^m}$ defined as:

$$H_\alpha(x_1, \dots, x_m) = \prod_{i=1}^{m} h_{\alpha_i}(x_i) \tag{37}$$

all the properties of the one-dimensional Hermite polynomials generalize to this case, in particular they form an orthogonal basis of $L^2(\mathbb{R}^m, \mathbb{Q})$ where $\mathbb{Q}$ is a multivariate normal distribution. If $\langle \cdot, \cdot \rangle$ is the inner product of that Hilbert space, we have that:

$$\langle H_\alpha, H_\beta \rangle = \alpha! \delta_{\alpha, \beta} \tag{38}$$

Of course all this is valid in the case $m = nd$, where we can also use multi-multi-indices to get the following identity:

$$H_{\underline{\alpha}}(x_1^1, \dots, x_d^n) = \prod_{\mu=1}^{n} H_{\alpha^\mu}(x^\mu) = \prod_{\mu=1}^{n} \prod_{i=1}^{d} h_{\alpha_i^\mu}(x_i^\mu) \tag{39}$$

We are now ready to see the proof of 75.

**Lemma 10.** *We consider a hypothesis testing problem in $\mathbb{R}^d$ where the null hypothesis $\mathbb{Q}$ is the multivariate Gaussian distribution $\mathcal{N}(0, \mathbb{1}_{d \times d})$, while the alternative hypothesis $\mathbb{P}$ is absolutely continuous with respect to it. Then:*

$$L^{\leq D} = \sum_{|\boldsymbol{\alpha}| \leq D} \frac{\langle L, H_{\boldsymbol{\alpha}} \rangle H_{\boldsymbol{\alpha}}}{\boldsymbol{\alpha}!} = \sum_{|\boldsymbol{\alpha}| \leq D} \frac{\mathbb{E}_{x \sim \mathbb{P}}[H_{\boldsymbol{\alpha}}(x)] H_{\boldsymbol{\alpha}}}{\boldsymbol{\alpha}!} \tag{40}$$

*Which implies*

$$||L^{\leq D}||^2 = \sum_{|\boldsymbol{\alpha}| \leq D} \frac{\langle L, H_{\boldsymbol{\alpha}} \rangle^2}{\boldsymbol{\alpha}!} = \sum_{|\boldsymbol{\alpha}| \leq D} \frac{\mathbb{E}_{x \sim \mathbb{P}}[H_{\boldsymbol{\alpha}}(x)]^2}{\boldsymbol{\alpha}!} \tag{41}$$

*Proof.* First note that

$$\langle L, H_{\boldsymbol{\alpha}} \rangle = \mathbb{E}_{x \sim \mathbb{P}}[H_{\boldsymbol{\alpha}}(x)] \tag{42}$$

due to the definition of likelihood ratio and a change of variable in the expectation. Then we can just use the fact that $(H_{\boldsymbol{\alpha}})_{\boldsymbol{\alpha}} \in \mathbb{N}^d$ are an orthogonal basis for $L^2(\mathbb{R}^d, \mathbb{Q})$, and if we consider the Hermite polynomials up to degree $D$, they are also a basis of the space of polynomials in which we want to project $L$ to get $L^{\leq D}$. Hence the formulas follow by just computing the projection using this basis. $\qquad \square$

Note that we set the lemma in $\mathbb{R}^d$, but of course it holds also in $\mathbb{R}^{nd}$ just switching to multi-multi-index notation.

### B.3.1   Hermite coefficients

Lemma 10 translates the problem of computing the norm of the LDLR to the problem of computing the projections $\langle L, H_{\boldsymbol{\alpha}} \rangle$. Note that this quantity is equal to $\mathbb{E}_{\mathbb{P}}[H_{\boldsymbol{\alpha}}(x)]$, which we will call $\boldsymbol{\alpha}$-*th Hermite coefficient* of the distribution $\mathbb{P}$.

The following lemma from [96] provides a version of the integration by parts technique that is tailored for Hermite polynomials.

**Lemma 11.** *Let $f : \mathbb{R}^d \to \mathbb{R}^d$ be a function that is continuously differentiable $k$ times. Assume that $f$ and all of its partial derivatives up to order $k$ are bounded by $O(\exp(|y|^\lambda))$ for some $\lambda \in (0, 2)$, then for any $\boldsymbol{\alpha} \in \mathbb{N}^d$ such that $|\boldsymbol{\alpha}| \leq k$*

$$\langle f, H_{\boldsymbol{\alpha}} \rangle = \mathbb{E}_{y \sim \mathcal{N}(0, \mathbb{1})} [H_{\boldsymbol{\alpha}}(y) f(y)] = \mathbb{E}_{y \sim \mathcal{N}(0, \mathbb{1})} [\partial_{\boldsymbol{\alpha}} f(y)] \tag{43}$$

*Proof.* 43 can be proved by doing induction on $k$ using 33, see [96] for details. $\qquad \square$

### B.3.2   Links with cumulant theory

The cumulants $(\kappa_{\boldsymbol{\alpha}})_{\boldsymbol{\alpha} \in \mathbb{N}^d}$ of a random variable $x \in \mathbb{R}^d$, can be defined as the Taylor coefficients of the expansion in $\xi$ of the *cumulant generating function*:

$$K_x(\xi) := \log \left( \mathbb{E} \left[ e^{\xi \cdot x} \right] \right) \tag{44}$$

Order-one cumulant is the mean, order-two is the variance, and higher-order cumulants cumulants encode more complex correlations among the variables. The Gaussian distribution is the only non constant distribution to have a polynomial cumulant generating function (as proved in theorem 7.3.5 in [100]), if $z \sim \mathcal{N}(\mu, \Sigma)$, then:

$$K_z(\xi) = \mu \xi + \frac{1}{2} \xi^\top \Sigma \xi.$$

Hence cumulants with order higher than three can also be seen as a measure of how much a distribution deviates from Gaussianity (i.e. how much it deviates from its best Gaussian approximation).

On this point the similarity with Hermite coefficients is evident. Indeed, up to order-five, on whitened distributions, cumulants and Hermite coefficients coincide. But form sixth-order onward, they start diverging. They are still linked by deterministic relationship, but its combinatorial complexity increases swiftly and it is not easy to translate formulas involving Hermite coefficients into cumulants

and vice versa. For this reason, our low-degree analysis of the likelihood ratio leading to 5 keeps the formalism that naturally arises from the computations, which is based on Hermite coefficients. A detailed discussion of the relation between Hermite polynomials and cumulants in the context of asymptotic expansions of distributions like the Gram–Charlier and Edgeworth series can be found in chapter 5 of McCullagh [97].

## B.4 Details on the spiked cumulant model

Here we will expand on the mathematical details of the spiked cumulant model.

### B.4.1 Prior distribution on the spike

For the prior distribution on $u$, $\mathcal{P}$, its role is analogous to the spiked Wishart model, so all the choices commonly used for that model can be considered applicable to this case. Namely symmetric distributions so that $\left\|\frac{u}{\sqrt{d}}\right\| \approx 1$ as $d \to \infty$. In the following we will make the computations assuming $u_i$ i.i.d. and with Rademacher prior:

$$u_i \sim \text{Rademacher}(1/2) \tag{45}$$

It helps to have i.i.d. components and constant norm $\left\|\frac{u}{\sqrt{d}}\right\| \equiv 1$. However all the results should hold, with more involved computations, also with the following priors:

- $(u_i)_{i=1,\ldots,d}$ are i.i.d. and $u_i \sim \mathcal{N}(0,1)$

- $u \in \text{Unif}(\partial B(0, \sqrt{d})$.

### B.4.2 Distribution of the non-Gaussianity

As detailed in 1, we need the non-Gaussianity to satisfy specific conditions. Some of the requirements are fundamental and cannot be avoided, whereas others could likely be removed or modified with only technical repercussion that do not change the essence of the results.

The most vital assumptions are the first and the last: $\mathbb{E}[g] \neq 0$ would introduce signal in the first cumulant, changing completely the model. It is important to control the tails of the distribution with

$$\mathbb{E}[\exp\left(g^2/2\right)] < +\infty, \tag{46}$$

a fat-tailed $g$ would make the LR-LDLR technique pointless due to the fact that $L \notin \mathcal{L}^2(\mathbb{R}^d, \mathbb{Q})$ and $\|L\| = \infty$ for any $n, d$. For example it is not possible to use Laplace distribution for $g$.

On the opposite side, the least important assumptions are that $\text{Var}(g) = 1$ and eq. (5); removing the first would just change the formula for whitening matrix $S$, while eq. (5) is a very weak requirement and it is needed just to estimate easily the Hermite coefficients' contribution and reach 11, it may be even possible to remove it and try to derive a similar estimate from the assumption on the tails 46.

Finally, the requirement of symmetry of the distribution has been chosen arbitrarily thinking about applications: the idea is that the magnitude of the fourth-order cumulant, the kurtosis, is sometimes used as a test for non-Gaussianity, so it is interesting to isolate the contribution given by the kurtosis, and higher-order, even degree, cumulants, by cancelling the contribution of odd-order cumulants. However, it would be interesting to extend the model in the case of a centered distribution with non-zero third-order cumulant. It is likely that the same techniques can be applied, but the final thresholds on $\theta$ may differ.

The following lemma gives a criterion of admissibility that ensures $\text{Radem}(1/2)$ and $\text{Unif}(-\sqrt{3}, \sqrt{3})$ (together with many other compactly supported distributions) satisfy 1.

**Lemma 12.** *Suppose $p_g$ is a probability distribution, compactly supported in $[-\Lambda, \Lambda]$, $\Lambda \geq 1$, then*

$$\mathbb{E}_{g \sim p_g} [h_m(g)] \leq \Lambda^m m!$$

*Proof.* Use the notation $h_m(x) = \sum_{k=0}^m a_{m,k} x^k$ and $S_m := \sum_{k=0}^m |a_{m,k}|$. Then we have that

$$
\begin{aligned}
\mathbb{E}_{g \sim p_g}[h_m(g)] &= \mathbb{E}_{g \sim p_g}\left[\sum_{k=0}^m a_{m,k} g^k\right] \\
&\leq \mathbb{E}_{g \sim p_g}\left[\sum_{k=0}^m |a_{m,k}||g|^k\right] \\
&\leq \Lambda^m S_m
\end{aligned}
\tag{47}
$$

So we just need to prove that $S_m \leq m!$, which can be done by induction using 34. Suppose it true for $m$, we prove it for $m + 1$. By 34 we have that:

$$
|a_{m+1,k}| \leq \begin{cases} |a_{m,1}| & k = 0 \\ |a_{m,k-1}| + (k+1)|a_{m,k+1}| & k > 0 \end{cases}
\tag{48}
$$

Summing on both sides (and using $a_{m,k} = 0$ when $k > m$), we get:

$$
\begin{aligned}
S_{m+1} &\leq |a_{m,1}| + \sum_{k=1}^{m+1} |a_{m,k-1}| + (k+1)|a_{m,k+1}| \\
S_{m+1} &\leq S_m + \sum_{j=0}^m j|a_{m,j}| \\
S_{m+1} &\leq S_m + m\sum_{j=0}^m |a_{m,j}| \\
S_{m+1} &\leq (m+1)S_m
\end{aligned}
\tag{49}
$$

Hence by application of the inductive hypothesis we get $S_{m+1} \leq (m+1)!$, completing the proof. $\square$

### B.4.3 Computing the whitening matrix

In this paragraph all the expectations are made assuming $u$ fixed and it will be best to work with its normalized version $\bar{u} = u/\sqrt{d}$. Note that $\mathbb{E}[x] = 0$ and we want also that

$$
\begin{aligned}
\mathbb{1}_{d \times d} = \mathbb{E}[xx^\top] &= \mathbb{E}[S\left(\sqrt{\beta}g\bar{u} + z\right)\left(\sqrt{\beta}g\bar{u}^\top + z^\top\right)S^\top] \\
&= S\left(\mathbb{1} + \beta\bar{u}\bar{u}^\top\right)S^\top
\end{aligned}
\tag{50}
$$

Hence we need

$$
S^2 = SS^T = \left(\mathbb{1} + \beta\bar{u}\bar{u}^\top\right)^{-1} = \mathbb{1} - \frac{\beta}{1+\beta}\bar{u}\bar{u}^\top
\tag{51}
$$

So we look for $\gamma$ such that:

$$
(\mathbb{1} + \gamma\bar{u}\bar{u}^\top)^2 = \mathbb{1} - \frac{\beta}{1+\beta}\bar{u}\bar{u}^\top
\tag{52}
$$

By solving the second-order equation we get

$$
\gamma_\pm = -\left(1 \pm \frac{1}{\sqrt{1+\beta}}\right)
\tag{53}
$$

We choose the solution with $-$, so that $S$ is positive definite:

$$
S = \mathbb{1} - \left(1 - \frac{1}{\sqrt{1+\beta}}\right)\bar{u}\bar{u}^\top = \mathbb{1} - \frac{\beta}{1+\beta+\sqrt{1+\beta}}\frac{uu^\top}{d}.
\tag{54}
$$

Hence we can compute also the explicit expression for $x$:

$$x = z - \left(1 - \frac{1}{\sqrt{1+\beta}}\right) \bar{u}^\top z \bar{u} + \sqrt{\frac{\beta}{1+\beta}} g \bar{u}$$

$$x = \underbrace{z - \bar{u}^\top z \bar{u}}_{z_{\perp u}} + \left(\sqrt{\frac{1}{1+\beta}} \bar{u}^\top z + \underbrace{\sqrt{\frac{\beta}{1+\beta}} g}_{\eta}\right) \bar{u}$$

$$= z_{\perp u} + \left(\sqrt{1 - \eta^2} \bar{u}^\top z + \eta g\right) \bar{u} \tag{55}$$

So $x$ is standard Gaussian in the directions orthogonal to $u$, whereas in $u$ direction, it is a weighted sum between $g$ and $z$. Note that it is a quadratic interpolation: the sum of the square of the weights is 1.

### B.5 Details of the LR analysis for spiked cumulant model

#### B.5.1 Proof sketch for theorem 2

Since the samples are independent, the total LR factorises as

$$L(\underline{y}) = \mathbb{E}_u \left[\prod_{\mu=1}^n l(y^\mu|u)\right], \tag{56}$$

where the sample-wise likelihood ratio is

$$l(y|u) = \frac{p_x(y|u)}{p_z(y)} = \mathbb{E}_{g \sim p_g}\left[\sqrt{1+\beta}\exp\left(-\frac{1+\beta}{2}\left(g - \sqrt{\frac{\beta}{(1+\beta)d}}y \cdot u\right)^2 + \frac{g^2}{2}\right)\right]. \tag{57}$$

To compute the norm of the LR, we consider two independent replicas of the spike, $u$ and $v$, to get

$$\|L_{n,d}\|^2 = \mathbb{E}_{\underline{y} \sim \mathbb{Q}}\left[\mathbb{E}_u\left[\prod_{\mu=1}^n l(y^\mu|u)\right]\mathbb{E}_v\left[\prod_{\mu=1}^n l(y^\mu|v)\right]\right]. \tag{58}$$

The hard part now is to simplify the high-dimensional integrals in this expression, it can be done because the integrand is almost completely symmetric, and the only asymmetries lie on the subspace spanned by $u$ and $v$.

$$\|L_{n,d}\|^2 = \mathbb{E}_{u,v}\left[\mathbb{E}_{g_u,g_v}\left[\frac{1+\beta}{\sqrt{(1+\beta)^2 - \beta^2\left(\frac{u \cdot v}{d}\right)^2}}e^{-\frac{(1+\beta)\left((1+\beta)(g_u^2+g_v^2) - 2\beta(g_u g_v)\left(\frac{u \cdot v}{d}\right)\right)}{2(1+\beta)^2 - 2\beta^2\left(\frac{u \cdot v}{d}\right)^2} + \frac{g_u^2+g_v^2}{2}}\right]^n\right]$$

Note that the integrand depends on $u$ and $v$ only trough $\frac{u \cdot v}{d} =: \lambda$. So, using that the prior on $u, v$ is i.i.d. Rademacher, the outer expectation can be transformed in a one dimensional expectation over $\lambda$, leading to (6).

#### B.5.2 Proof of theorem 2

$\tilde{x}^\mu = \sqrt{\frac{\beta}{d}}g^\mu u + z^\mu$, to find the marginal density of $x^\mu$ we integrate over the possible values of $g$

$$p_{\tilde{x}}(y|u) = \mathbb{P}\left(\tilde{x}^\mu \in \mathrm{d}y|u\right) = \mathbb{E}_g\left[p_z\left(y - \sqrt{\frac{\beta}{d}}gu\right)\right] \tag{59}$$

where $p_z$ is the density of a standard normal $d$-dimensional variable $z \sim \mathcal{N}(0, \mathbb{1}_d)$. But we are interested in the density of the whitened variable $x$, $p_x(\cdot|u)$, which can be seen as the push forward of the density of $\tilde{x}$ with respect to the linear transformation $S$. So

$$p_x(y|u) = p_{\tilde{x}}(S^{-1}y|u)|\det S^{-1}|. \tag{60}$$

It is easy to see from 3 that $|\det S^{-1}| = \sqrt{1+\beta}$, so we can plug it into 59 and after expanding the computations we get to

$$p_x(y|u) = p_z(y)\mathbb{E}_g\left[\sqrt{1+\beta}\exp\left(-\frac{1+\beta}{2}\left(g-\sqrt{\frac{\beta}{(1+\beta)d}}y\cdot u\right)^2 + \frac{g^2}{2}\right)\right]. \quad (61)$$

So we have found the likelihood ratio for a single sample, conditioned on the spike:

$$l(y|u) = \frac{p_x(y|u)}{p_z(y)} = \mathbb{E}_g\left[\sqrt{1+\beta}\exp\left(-\frac{1+\beta}{2}\left(g-\sqrt{\frac{\beta}{(1+\beta)d}}y\cdot u\right)^2 + \frac{g^2}{2}\right)\right]. \quad (62)$$

Note that conditioning on $u$ the samples are independent, so we have the following formula:

$$L(\underline{y}) = \mathbb{E}_u\left[\prod_{\mu=1}^n l(y^\mu|u)\right]. \quad (63)$$

So to compute the norm we consider two independent replicas of the spike, $u$ and $v$, then we switch the order of integration to get

$$\begin{aligned}
||L_{n,d}||^2 &= \mathbb{E}_{y\sim\mathbb{Q}}\left[\mathbb{E}_u\left[\prod_{\mu=1}^n l(y^\mu|u)\right]\mathbb{E}_v\left[\prod_{\mu=1}^n l(y^\mu|v)\right]\right] \\
&= \mathbb{E}_{u,v}\left[\mathbb{E}_{y\sim\mathbb{Q}}\left[\prod_{\mu=1}^n l(y^\mu|u)l(y^\mu|v)\right]\right] \\
&= \mathbb{E}_{u,v}\left[\mathbb{E}_{y\sim\mathbb{Q}}\left[l(y|u)l(y|v)\right]^n\right] \\
&= \mathbb{E}_{u,v}\left[\mathbb{E}_{y\sim\mathbb{Q}}\left(\mathbb{E}_{g_u}\left[\sqrt{1+\beta}\exp\left(-\frac{1+\beta}{2}\left(g_u-\sqrt{\frac{\beta}{(1+\beta)d}}y\cdot u\right)^2 + \frac{g_u^2}{2}\right)\right]\right.\right. \\
&\qquad\left.\left.\cdot\mathbb{E}_{g_v}\left[\sqrt{1+\beta}\exp\left(-\frac{1+\beta}{2}\left(g_v-\sqrt{\frac{\beta}{(1+\beta)d}}y\cdot v\right)^2 + \frac{g_v^2}{2}\right)\right]\right)^n\right].
\end{aligned} \quad (64)$$

Switching the integral over $y$ inside the new integrals over $g_u$ and $g_v$ we can isolate the following integral over $y$:

$$I := \mathbb{E}_{y\sim\mathbb{Q}}\left[\exp\left(-\frac{1+\beta}{2}\left(g_u-\sqrt{\frac{\beta}{(1+\beta)d}}y\cdot u\right)^2 - \frac{1+\beta}{2}\left(g_v-\sqrt{\frac{\beta}{(1+\beta)d}}y\cdot v\right)^2\right)\right]. \quad (65)$$

It can be computed by noting the subspace orthogonal to $\{u,v\}$, we just have the integral of a standard normal, and the remaining 2-dimensional integral can be computed explicitly. Since the Rademacher prior implies that $||u|| = ||v|| = \sqrt{d}$, the result depends only on their overlap $\lambda$ (i.e. $\lambda = \frac{u\cdot v}{d}$), leading to:

$$I = \frac{1}{\sqrt{(1+\beta)^2 - \beta^2\lambda^2}}\exp\left(-\frac{1+\beta}{2(1+\beta)^2 - 2\beta^2\lambda^2}\left((1+\beta)(g_u^2+g_v^2) - 2\beta(g_ug_v)\lambda\right)\right) \quad (66)$$

If we plug this formula inside 64 and rearrange the terms we get an expression that can be written in terms of the density of two bi-dimensional centered Gaussians

$$\begin{aligned}
||L_{n,d}||^2 &= \mathbb{E}_\lambda\left[\mathbb{E}_{g_u,g_v}\left[\frac{1+\beta}{\sqrt{(1+\beta)^2 - \beta^2\lambda^2}}e^{-\frac{(1+\beta)\left((1+\beta)(g_u^2+g_v^2)-2\beta(g_ug_v)\lambda\right)}{2(1+\beta)^2 - 2\beta^2\lambda^2} + \frac{g_u^2+g_v^2}{2}}\right]^n\right] \\
&= \mathbb{E}_\lambda\left[\mathbb{E}_{g_u,g_v}\left[\mathcal{N}\left((g_u,g_v);\Sigma\right)\mathcal{N}\left((g_u,g_v);\mathbb{1}_{2\times2}\right)^{-1}\right]^n\right].
\end{aligned} \quad (67)$$

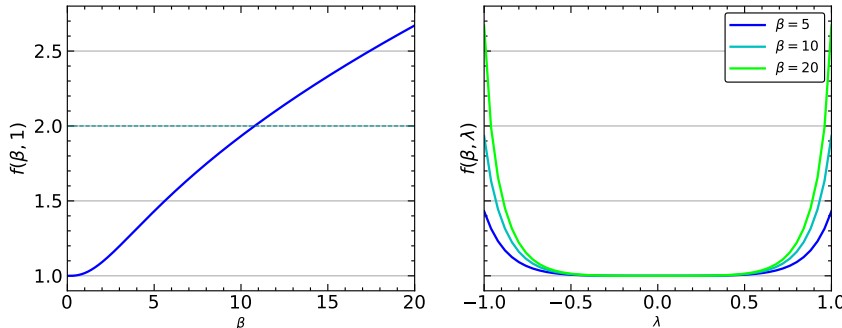

Figure 6: Graphs of $f$, defined in (7), when $g \sim \text{Rademacher}(1/2)$.

where

$$\Sigma^{-1} = \frac{1+\beta}{(1+\beta)^2 - \beta^2\lambda^2} \begin{pmatrix} 1+\beta & -\beta\lambda \\ -\beta\lambda & 1+\beta \end{pmatrix}, \qquad \lambda = \frac{u \cdot v}{d}. \tag{68}$$

Now we turn to compute the expectation over $\lambda$. Note that since $u, v$ are independent and their components are Rademacher distributed, product of Rademacher is still Rademacher, hence $u \cdot v$ is the sum of $d$ independent Rademacher random variables. Using moreover that the Rademacher distribution is just a linear transformation of the Bernoulli, we can link the distribution of a the overlap $\lambda$ to a linear transformation of a binomial distribution: $\frac{d}{2}(\lambda + 1) \sim \text{Binom}(d, 1/2)$.

We therefore define the auxiliary function

$$f(\beta, \lambda) := \mathop{\mathbb{E}}_{g_u, g_v} \left[ \frac{1+\beta}{\sqrt{(1+\beta)^2 - \beta^2\lambda^2}} e^{-\frac{(1+\beta)\left((1+\beta)(g_u^2 + g_v^2) - 2\beta(g_u g_v)\lambda\right)}{2(1+\beta)^2 - 2\beta^2\lambda^2} + \frac{g_u^2 + g_v^2}{2}} \right] \tag{69}$$

so that the LR norm can be rewritten as

$$||L_{n,d}||^2 = \sum_{j=0}^{d} \binom{d}{j} \frac{1}{2^d} f\left(\beta, \frac{2j}{d} - 1\right)^n \tag{70}$$

which is what we needed to prove.

### B.5.3 Consequences of theorem 2

The precise value of $f$ depends on the choice of distribution for the non-Gaussianity $g$, however, it is possible to prove the following

**Lemma 13.** *If $p_g$ satisfies assumptions 1 and $\theta < 1$, then there exists $C$ such that:*

$$||L_{n,d}|| \leq C \qquad \forall n, d$$

*Proof.* We first prove that, thanks to the sub-Gaussianity of $g$, we have that

$$f(\beta, \lambda) \leq 1 + C_\beta \lambda^2, \qquad \forall \lambda \in [-1, 1] \tag{71}$$

To do it, first note that $f\beta, 0) = 1$, and $f(\beta, \cdot)$ is bounded on $[-1, 1]$ thanks to the sub-gaussianity of $g$:

$$|f(\beta, \lambda)| \leq \mathbb{E}_{g_u, g_v} \left[ e^{\frac{1}{2}\left(g_u^2 + g_v^2\right)} \right] \sup_{g_u, g_v} \left( \frac{1+\beta}{\sqrt{(1+\beta)^2 - \beta^2\lambda^2}} e^{-\frac{(1+\beta)\left((1+\beta)(g_u^2 + g_v^2) - 2\beta(g_u g_v)\lambda\right)}{2(1+\beta)^2 - 2\beta^2\lambda^2}} \right)$$

$$\leq C \frac{1+\beta}{\sqrt{(1+\beta)^2 - \beta^2\lambda^2}}$$

$$\leq C_1 \left(1 + C_2 \lambda^2\right)$$

So we just need to prove that, up to changes on $C_2$, we can take $C_1 = 1$. To do that it si sufficient to study the behaviour around $\lambda = 0$:

$$\frac{\partial}{\partial\lambda} f(\beta, 0) = \mathbb{E}\left[ \frac{\beta}{1+\beta} g_u g_v \right] = 0$$

So there is a neighboorood of 0 such that $f(\beta, 0) = 1 + C\lambda^2 + o(\lambda^2)$. Hence we can take a suitable constant $C_\beta$ so that eq. (71) holds.

Now we can apply eq. (71) to (6), to get

$$||L_{n,d}||^2 \leq \sum_{j=0}^{d} \binom{d}{j} \frac{1}{2^d} \left( 1 + C_\beta \left( \frac{2j}{d} - 1 \right)^2 \right)^n$$

$$= \sum_{j=0}^{d} \binom{d}{j} \frac{1}{2^d} \sum_{k=0}^{n} \binom{n}{k} C_\beta^k \left( \frac{2j}{d} - 1 \right)^{2k}$$

$$= \sum_{k=0}^{n} \binom{n}{k} \left( \frac{C_\beta}{d^2} \right)^k \mathbb{E}\left[ Y_d^{2k} \right]$$

Where $Y_d$ is a random variable distributed as the sum of $d$ independent Rademacher with parameter $1/2$. So we can apply the central limit theorem on $\frac{Y_d}{\sqrt{d}}$ to get that $\frac{\mathbb{E}[Y_d^{2k}]}{d} \to (2k-1)!!$ hence we get:

$$||L_{n,d}||^2 \leq C \sum_{k=0}^{n} \binom{n}{k} \left( \frac{C_\beta}{d} \right)^k (2k-1)!!$$

$$\leq C_1 \sum_{k=0}^{n} \left( \frac{C_2 n}{d} \right)^k$$

where $C_1$ and $C_2$ are constants independent of $n, d$. So now we can substitute that $n \asymp d^\theta$, and since $\theta < 1$ the series converges for all $d$, hence the LR norm is bounded. $\qquad\square$

We will analyze in more detail the case in which $g \sim \text{Rademacher}(1/2)$. It is a case that is particularly interesting for the point of view of the applications because it amounts to comparing a standard Gaussian with a Gaussian mixture with same mean and covariance. Moreover, with this choice of non-Gaussianity, the technical work simplifies because the troublesome integral over $g_u, g_v$ in 7 becomes just a simple sum over 4 possibilities. In this case $f$ can be computed exactly and it is displayed in 6. The maximum of $f(\beta, \cdot)$ is attained at $\lambda = \pm 1$ and $f(\cdot, 1)$ is monotonically increasing.

Assume that $n \asymp d^\theta$, then a sufficient condition for LR to diverge is $\frac{f(\beta,1)^{d^\theta}}{2^d} \to \infty$, which holds as soon as $\theta > 1$. Moreover, even at linear sample complexity, it is possible to find regimes that ensure divergence of the LR norm, similar to BBP phase transition in spiked Wishart model. Assume that samples and dimensions scale at the same rate as in spiked Wishart model: $n \asymp \frac{d}{\gamma}$, then a sufficient condition for divergence is that

$$\frac{f(\beta, 1)^{d/\gamma}}{2^d} \to \infty$$

which holds if and only if $f(\beta, 1)^{1/\gamma} > 2$. Hence given $\beta$, you can always find

$$\gamma_\beta := \frac{\log \left( f(\beta, 1) \right)}{\log 2} \tag{72}$$

And for $\gamma > \gamma_\beta$ there is guarantee that $||L_{n,d}|| \to \infty$. Vice-versa, we could also fix $\gamma$ and define $\beta_\gamma$ as only value of $\beta$ that makes 72 true.

It is spontaneous to ask whether this condition for divergence of LR norm is also necessary, making the SNR threshold $\beta_\gamma$ the analogous in this model of the threshold $\beta_c = \sqrt{\gamma}$ in spiked Wishart model (8). Although a rigorous proof of this is still missing, numerical evidence (7) suggests that for $\beta \leq \beta_\gamma$ the LR norm stays indeed bounded.

## B.6 LDLR on spiked cumulant model

To discuss the proof of theorem 5, it is best to state it in two separate parts

**Theorem 14** (LDLR for spiked cumulant model)**.** *Suppose that $(u_i)_{i=1,...,d}$ are drawn i.i.d. from the symmetric Rademacher distribution. If the non-Gaussian distribution $p_g$ satisfies assumption 1, then the following lower and upper bounds hold:*

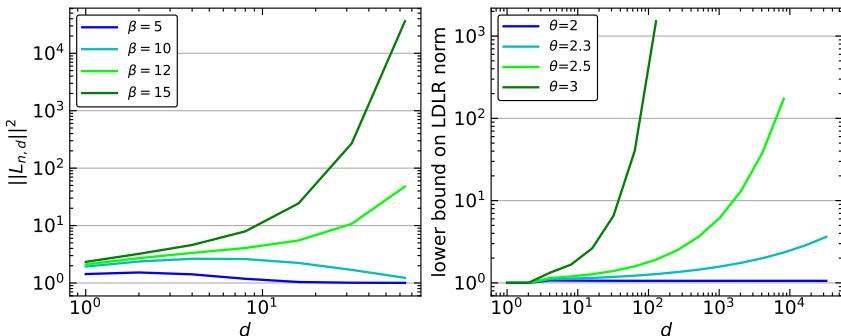

Figure 7: On the left, LR norm when $g \sim$ Rademacher$(1/2)$ in the regime $n = \gamma d$ with $\gamma = 1$. When $\beta < \beta_\gamma \approx 10.7$ the likelihood ratio remains bounded, whereas it goes to $+\infty$ for $\beta > \beta_\gamma$. On the right, the lower bound on $||L_{n,d}^{\leq D(n)}||$ given by 73 goes to $+\infty$ for $\theta > 2$. Parameters for the plot: $g \sim$ Radem$(1/2)$, $\beta = 10$, $D(n) = \log^{3/2}(n)$

- *Let $D \in \mathbb{N}$ such that $D/4 \leq n$, then:*

$$\left\| L_{n,d}^{\leq D} \right\|^2 \geq \sum_{m=0}^{\lfloor D/4 \rfloor} \binom{n}{m} \binom{d+1}{2}^m \left( \frac{\beta^2 \kappa_4^g}{\sqrt{4!}d^2(1+\beta)^2} \right)^{2m}. \tag{73}$$

  *where $\kappa_4^g$ is the fourth-order cumulant of $g$.*

- *Conversely, for any $D, n, d$:*

$$\left\| L_{n,d}^{\leq D} \right\|^2 \leq 1 + \sum_{m=1}^{D} \frac{C_m}{d^m} \sup_{k \leq m} \left( \mathbb{E}\left[ h_k(g) \right]^{2m/k} \right) \binom{n}{\lfloor m/4 \rfloor} \binom{d}{\lfloor m/2 \rfloor} \tag{74}$$

  *where $C_m := \left( \frac{\beta}{1+\beta} \right)^m \binom{\lfloor m/4 \rfloor \lfloor m/2 \rfloor + m - 1}{m}$.*

**Corollary 15** (Asymptotics of LDLR bounds). *Assume the hypotheses of theorem 5. Let $0 < \varepsilon < 1$ and assume $D(n) \asymp \log^{1+\varepsilon}(n)$. Take $n, d \to \infty$, with the scaling $n \asymp d^\theta$ for $\theta > 0$. Estimate (73) implies that for $n, d$ large enough, the following lower bound holds:*

$$\left\| L_{n,d}^{\leq D(n)} \right\|^2 \geq \left( \frac{1}{\lfloor D(n)/4 \rfloor} \left( \frac{\beta^2 \kappa_4^g}{(1+\beta)^2} \right)^2 \frac{n}{d^2} \right)^{\lfloor D(n)/4 \rfloor}$$

*Conversely, (74) leads to:*

$$\left\| L_{n,d}^{\leq D(n)} \right\|^2 \leq 1 + \sum_{m=1}^{D(n)} \left( \frac{\Lambda^2 \beta}{1+\beta} \right)^m m^{4m} \left( \frac{n}{d^2} \right)^{m/4}$$

*Taken together, (10) and (11) imply the presence of a critical regime for $\theta_c = 2$, and describe the behaviour of $\left\| L_{n,d}^{\leq D} \right\|$ for all $\theta \neq \theta_c$*

$$\lim_{n,d \to \infty} \left\| L_{n,d}^{\leq D(n)} \right\| = \begin{cases} 1 & 0 < \theta < 2 \\ +\infty & \theta > 2 \end{cases}$$

In the following we will present the proofs of 14 and 15. We first give a sketch of the proof in B.6.1 before giving the detailed proof in B.6.2.

### B.6.1 Proof sketch

The starting point of the argument is the observation that since the null hypothesis is white Gaussian noise, the space $L^2(\mathbb{R}^{nd}, \langle \cdot, \cdot \rangle)$ has an explicit orthogonal basis in the set of multivariate Hermite

polynomials $(H_{\boldsymbol{\alpha}})$. The multi-index $\boldsymbol{\alpha} \in \mathbb{N}^{nd}$ denotes the degree of the Hermite polynomial which is computed for each entry of the data matrix; see B.3 for a detailed explanation. We can then expand the LDLR norm in this basis to write:

$$\|L_{n,d}^{\leq D}\|^2 = \sum_{|\boldsymbol{\alpha}| \leq D} \frac{\langle L, H_{\boldsymbol{\alpha}} \rangle^2}{\boldsymbol{\alpha}!} = \sum_{|\boldsymbol{\alpha}| \leq D} \frac{1}{\boldsymbol{\alpha}!} \underset{x \sim \mathbb{P}}{\mathbb{E}}[H_{\boldsymbol{\alpha}}(x)]^2 \tag{75}$$

From now on the two bounds are treated separately.

**Lower bound** The idea is to bound the LDLR from below by computing the sum 75 only over a restricted set of addends $\mathcal{I}_m$ which we can compute explicitly. In particular, we consider the set $\mathcal{I}_m$ of all the polynomials with degree $4m$ in the data matrix, which are at most of order 4 in each individual sample $x^\mu$. Then we use that the expectation of such Hermite polynomials conditioned on $u$ can be split into $m$ integrals of degree-4 Hermite polynomials in $d$ variables. In this way we can use our knowledge of the fourth-order cumulant of $x$ to compute those expectations (see 93), so we find that

$$\|L_{n,d}^{\leq D}\|^2 \geq \sum_m^{\lfloor D/4 \rfloor} \sum_{|\boldsymbol{\alpha}| \in \mathcal{I}_m} \frac{1}{\boldsymbol{\alpha}!} \underset{x \sim \mathbb{P}}{\mathbb{E}}[H_{\boldsymbol{\alpha}}(x)]^2 \geq \sum_m^{\lfloor D/4 \rfloor} \sum_{\boldsymbol{\alpha} \in \mathcal{I}_m} \left( \frac{\beta^2 \kappa_4^g}{\sqrt{4!} d^2 (1+\beta)^2} \right)^{2m}. \tag{76}$$

Thanks to this manipulation, the bound does not depend on $\underline{\boldsymbol{\alpha}}$ explicitly, so we can complete the bound by estimating the cardinality of $\mathcal{I}_m$. Thanks to the fact that we have $n$ i.i.d. copies of variable $x$ at disposal, the class of polynomials $\mathcal{I}_m$ has size that grows at least as $\binom{n}{m}\binom{d+1}{2}^m$, allowing to reach the lower bound in the statement.

**Upper bound** We can show that the expectation of Hermite polynomials for a single sample $x^\mu$ over the distribution $\mathbb{P}(\cdot|u)$ can be written as $\mathbb{E}_{x \sim \mathbb{P}(\cdot|u)}[H_{\boldsymbol{\alpha}}(x)] = T_{|\boldsymbol{\alpha}|,g}/d^{|\boldsymbol{\alpha}|/2} u^{\boldsymbol{\alpha}}$, where $T_{|\boldsymbol{\alpha}|,g} = \left( \frac{\beta}{1+\beta} \right)^{|\boldsymbol{\alpha}|/2} \mathbb{E}\left[ h_{|\boldsymbol{\alpha}|}(g) \right]$, cf. lemma 16. Using the fact that inputs are sampled i.i.d. from $\mathbb{P}(\cdot|u)$ and substituting this result into 75, we find that

$$\|L_{n,d}^{\leq D}\|^2 = \sum_{|\underline{\boldsymbol{\alpha}}| \leq D} \frac{\left( \prod_{\mu=1}^n T_{|\boldsymbol{\alpha}^\mu|,g} \right)^2}{\underline{\boldsymbol{\alpha}}! d^{|\underline{\boldsymbol{\alpha}}|}} \underset{u \sim \mathcal{P}(u)}{\mathbb{E}}[u^{\underline{\boldsymbol{\alpha}}}]^2. \tag{77}$$

To obtain an upper bound, we will first show that many addends in this sum are equal to zero, then estimate the remainder.

On the one hand, since we chose a Rademacher prior over the elements of the spike $u_i$, the expectation over the prior yields either 1 or 0 depending on whether the exponent of each $u_i$ is even or odd. On the other hand, the whitening of the data means that $T_{|\boldsymbol{\alpha}|,g} = 0$ for $0 < |\boldsymbol{\alpha}| < 4$ (as proved in lemma 16).

For each $m \in \mathbb{N}$, we can denote $\mathcal{A}_m$ the set of multi-indices $|\underline{\boldsymbol{\alpha}}| = m$ that give non-zero contributions in 77, so that we can write:

$$\|L_{n,d}^{\leq D}\|^2 = \sum_{m=0}^D \sum_{\underline{\boldsymbol{\alpha}} \in \mathcal{A}_m} \frac{\left( \prod_{\mu=1}^n T_{|\boldsymbol{\alpha}^\mu|,g} \right)^2}{\underline{\boldsymbol{\alpha}}! d^m} \tag{78}$$

Now we proved that we can bound the inner terms so that they depend on $\underline{\boldsymbol{\alpha}}$ only through the norm $|\underline{\boldsymbol{\alpha}}| = m$ (cf. 82 and 110):

$$\|L_{n,d}^{\leq D}\|^2 \leq \sum_{m=0}^D \sum_{\underline{\boldsymbol{\alpha}} \in \mathcal{A}_m} \frac{1}{d^m} \left( \frac{\beta}{1+\beta} \right)^m \sup_{k \leq m} \left( \mathbb{E}[h_k(g)]^{2m/k} \right)$$

$$= \sum_{m=0}^D \frac{1}{d^m} \left( \frac{\beta}{1+\beta} \right)^m \sup_{k \leq m} \left( \mathbb{E}[h_k(g)]^{2m/k} \right) \#\mathcal{A}_m \tag{79}$$

Finally, the cardinality of $\mathcal{A}_m$ is 1 in case $m = 0$ (which leads to the addend 1 in 74) and if $m > 0$ it can be bounded by (for details see 111 in the appendix)

$$\binom{\lfloor m/4 \rfloor \lfloor m/2 \rfloor + m - 1}{m} \binom{n}{\lfloor m/4 \rfloor} \binom{d}{\lfloor m/2 \rfloor}, \tag{80}$$

leading to (74).

### B.6.2 Detailed proof

First we prove a lemma that provides formulas and estimates for projections of the sample-wise likelihood ratio $l(\cdot | u)$ on the Hermite polynomials.

**Lemma 16.** *Let* $x = S\left(\sqrt{\beta/d} g u + z\right)$ *be a spiked cumulant random variable. Then for any* $\boldsymbol{\alpha} \in \mathbb{N}^d$, *with* $|\boldsymbol{\alpha}| = m$, *we have that:*

$$\langle l(\cdot | u), H_{\boldsymbol{\alpha}} \rangle = \mathop{\mathbb{E}}_{x \sim \mathbb{P}(\cdot | u)} [H_{\boldsymbol{\alpha}}(x)] = \frac{T_{m,g}}{d^{m/2}} u^{\boldsymbol{\alpha}} \tag{81}$$

*where* $T_{m,g}$ *is a coefficient defined as:*

$$T_{m,g} = \left(\frac{\beta}{1+\beta}\right)^{m/2} \mathbb{E}[h_m(g)] \tag{82}$$

*Proof.* Recall that by (57)

$$l(y | u) = \mathop{\mathbb{E}}_{g}\left[\sqrt{1+\beta} \exp\left(-\frac{1}{2}\left(\sqrt{1+\beta} g - \sqrt{\frac{\beta}{d}} y \cdot u\right)^2 + \frac{g^2}{2}\right)\right]. \tag{83}$$

Note that this expression, thanks to 46 that bounds the integral, is differentiable infinitely many times in the $y$ variable. It can be quickly proven by induction on $|\boldsymbol{\alpha}| = m$ (using the recursive definition of Hermite polynomials 33) that:

$$\partial_{\boldsymbol{\alpha}} l(y | u) = \left(\frac{\beta}{d}\right)^{m/2} u^{\boldsymbol{\alpha}}.$$

$$\cdot \mathop{\mathbb{E}}_{g}\left[\sqrt{1+\beta}\, h_m\left(\sqrt{1+\beta} g - \sqrt{\frac{\beta}{d}} y \cdot u\right) \exp\left(-\frac{1}{2}\left(\sqrt{1+\beta} g - \sqrt{\frac{\beta}{d}} y \cdot u\right)^2 + \frac{g^2}{2}\right)\right] \tag{84}$$

Hence we can again use 46 together with the fact that

$$\sup_{g}\left| h_m\left(\sqrt{1+\beta} g - \sqrt{\frac{\beta}{d}} y \cdot u\right) \exp\left(-\frac{1}{2}\left(\sqrt{1+\beta} g - \sqrt{\frac{\beta}{d}} y \cdot u\right)^2\right)\right| < \infty$$

to deduce that the hypothesis of lemma 11 are met for any $\boldsymbol{\alpha} \in \mathbb{N}^d$, leading to:

$$\langle l(\cdot | u), H_{\boldsymbol{\alpha}} \rangle = \mathop{\mathbb{E}}_{x \sim \mathbb{P}(\cdot | u)} [H_{\boldsymbol{\alpha}}(x)] = \mathop{\mathbb{E}}_{y \sim \mathbb{Q}} [\partial_{\boldsymbol{\alpha}} l(y | u)] \tag{85}$$

This, and 84 already prove 81. Now to compute the exact value of $T_{m,g}$ we need to take the expectation with respect to $y \sim \mathbb{Q} = \mathcal{N}(0, \mathbb{1}_{d \times d})$. Note that by the choice of Rademacher prior on $u$, we know that $\|u\| = \sqrt{d}$. So, conditioning on $u$, $y \cdot \frac{u}{\sqrt{d}} \sim \mathcal{N}(0,1)$. Hence switching the expectations in 84, we get:

$$T_{m,g} = \beta^{m/2} \mathop{\mathbb{E}}_{g}\left[\int_{-\infty}^{\infty} \frac{\mathrm{d}z}{\sqrt{2\pi}} \sqrt{1+\beta}\, h_m\left(\sqrt{1+\beta} g - \sqrt{\beta} z\right) e^{-\frac{1}{2}\left(\sqrt{1+\beta} g - \sqrt{\beta} z\right)^2 + \frac{g^2 - z^2}{2}}\right]$$

$$= \beta^{m/2} \mathop{\mathbb{E}}_{g}\left[\int_{-\infty}^{\infty} \frac{\mathrm{d}z}{\sqrt{2\pi}} \sqrt{1+\beta}\, h_m\left(\sqrt{1+\beta} g - \sqrt{\beta} z\right) \exp\left(-\frac{1}{2}\left(\sqrt{\beta} g - \sqrt{1+\beta} z\right)^2\right)\right]$$

$$\stackrel{\tilde{z}=\sqrt{1+\beta} z}{=} \beta^{m/2} \mathop{\mathbb{E}}_{g}\left[\int_{-\infty}^{\infty} \frac{\mathrm{d}\tilde{z}}{\sqrt{2\pi}}\, h_m\left(\sqrt{1+\beta} g - \sqrt{\frac{\beta}{1+\beta}} \tilde{z}\right) \exp\left(-\frac{1}{2}\left(\sqrt{\beta} g - \tilde{z}\right)^2\right)\right] \tag{86}$$

Now we use 35 on

$$x = \frac{\beta}{\sqrt{1+\beta}} g - \sqrt{\frac{\beta}{1+\beta}} \tilde{z}$$

$$y = \sqrt{1+\beta} g - \frac{\beta}{\sqrt{1+\beta}} g = \frac{g}{\sqrt{1+\beta}}$$

(87)

applying also the translation change of variable $\hat{z} = \tilde{z} - \sqrt{\beta} g$ we get:

$$\mathop{\mathbb{E}}_{y\sim\mathbb{Q}} [\partial_{\boldsymbol{\alpha}} l(y|u)] = \left(\frac{\beta}{d}\right)^{m/2} u^{\boldsymbol{\alpha}} \mathop{\mathbb{E}}_{g} \left[ \sum_{k=0}^{m} \binom{m}{k} h_k \left(\frac{g}{\sqrt{1+\beta}}\right) \left(-\sqrt{\frac{\beta}{1+\beta}}\right)^{m-k} \mathop{\mathbb{E}}_{\hat{z}\sim\mathcal{N}(0,1)} [\hat{z}^{m-k}] \right]$$

(88)

Now recall the formula for the moments of the standard Gaussian (see for instance section 3.9 in [97]):

$$\mathop{\mathbb{E}}_{\hat{z}\sim\mathcal{N}(0,1)} [\hat{z}^{m-k}] = \begin{cases} (m-k-1)!! & \text{if } m-k \text{ is even} \\ 0 & \text{if } m-k \text{ is odd} \end{cases}$$

(89)

Plugging this formula inside and changing the summation index $2j := m - k$ we get

$$T_{m,g} = \beta^{m/2} \sum_{j=0}^{\lfloor m/2 \rfloor} \binom{m}{2j} \left(\frac{\beta}{1+\beta}\right)^j (2j-1)!! \mathbb{E}\left[ h_{m-2j} \left(\frac{g}{\sqrt{1+\beta}}\right) \right].$$

(90)

Note that 36 with $x = \frac{g}{\sqrt{1+\beta}}$ and $\gamma = \sqrt{1+\beta}$ gives the following rewriting of $h_m(g)$:

$$h_m(g) = \sum_{j=0}^{\lfloor m/2 \rfloor} (\sqrt{1-\beta})^{m-2j} (\beta)^j \binom{m}{2j} (2j-1)!! h_{m-2j} \left(\frac{g}{\sqrt{1+\beta}}\right)$$

(91)

which is almost the same expression as in 90. This allows simplify everything, leading to 82. $\square$

Note that lemma 16 together with 1 on $g$ imply:

$$m = 2 \text{ or } m \text{ odd} \implies T_{m,g} = 0,$$

(92)

So, apart from $m = 0$ which gives the trivial contribution $+1$, the first non zero contributions to the LDLR norm is at degree $m = 4$:

$$\mathop{\mathbb{E}}_{x\sim\mathbb{P}(\cdot|u)} [H_{\boldsymbol{\alpha}}(x)] = \frac{\beta^2}{(1+\beta)^2} \kappa_4^g \frac{u^{\boldsymbol{\alpha}}}{d^2}$$

(93)

From now on we will consider separately the lower and the upper bounds.

**Lower bound**  The idea of the proof is to start from

$$\|L_{n,d}^{\le D}\|^2 = \sum_{\substack{\boldsymbol{\alpha}\in\mathbb{N}^{nd} \\ |\boldsymbol{\alpha}|\le D}} \frac{\langle L, H_{\boldsymbol{\alpha}}\rangle^2}{\boldsymbol{\alpha}!}$$

(94)

and to estimate it from below by picking only few terms in the sum. So we restrict to particular sets of multi-multi-indices for which we can exploit 93. Let $m \in \mathbb{N}$ such that $4m \le D$, define

$$\mathcal{I}_m = \left\{ \underline{\boldsymbol{\alpha}} \in \left(\mathbb{N}^d\right)^n \, \middle| \, |\underline{\boldsymbol{\alpha}}| = 4m, \ |\boldsymbol{\alpha}^{\mu}| \in \{0,4\} \ \forall \ 1 \le \mu \le n \right\}$$

(95)

$\mathcal{I}_m$ is non empty since $m < n$ and for each $\underline{\boldsymbol{\alpha}} \in \mathcal{I}_m$ we can enumerate all the indices $\mu_1, \ldots, \mu_m$ such that $\boldsymbol{\alpha}^{\mu_i} \ne 0$.

Now we go on to compute the term $\langle L, H_{\underline{\boldsymbol{\alpha}}}\rangle$ for $\underline{\boldsymbol{\alpha}} \in \mathcal{I}_m$:

$$\langle L, H_{\underline{\boldsymbol{\alpha}}}\rangle = \int_{\mathcal{U}} \mathop{\mathbb{E}}_{\underline{x}\sim\mathbb{P}(\cdot|u)^{\otimes n}} [H_{\underline{\boldsymbol{\alpha}}}(\underline{x})] d\mathcal{P}(u)$$

$$= \int_{\mathcal{U}} \mathop{\mathbb{E}}_{\underline{x}\sim\mathbb{P}(\cdot|u)^{\otimes n}} \left[ \prod_{i=1}^{m} H_{\boldsymbol{\alpha}^{\mu_i}}(x^{\mu_i}) \right] d\mathcal{P}(u)$$

(96)

Since the samples are independent conditionally on $u$, we can split the inner expectation along the $m$ contributing directions:

$$\langle L, H_{\underline{\alpha}} \rangle = \int_{\mathcal{U}} \prod_{i=1}^{m} \mathbb{E}_{x \sim \mathbb{P}(\cdot|u)} [H_{\boldsymbol{\alpha}^{\mu_i}}(x^{\mu_i})] \mathrm{d}\mathcal{P}(u) \tag{97}$$

Now we need to compute the inner $d$-dimensional expectation. For that we use that $|\boldsymbol{\alpha}^{\mu_i}| = 4$, so, recalling the notation $\eta = \sqrt{\frac{\beta}{1+\beta}}$, we can apply 93 to get for each $i$

$$\mathbb{E}_{x \sim \mathbb{P}(\cdot|u)} [H_{\boldsymbol{\alpha}^{\mu_i}}(x^{\mu_i})] = \frac{\eta^4}{d^2} \kappa_4^g u^{\alpha^{\mu_i}}. \tag{98}$$

So the resulting integral can be written in the following way:

$$\langle L, H_{\underline{\alpha}} \rangle = \left( \frac{\eta^4 \kappa_4^g}{d^2} \right)^m \int_{\mathcal{U}} \prod_{j=1}^{d} u_j^{\gamma_j} \mathrm{d}\mathcal{P}(u) \tag{99}$$

where for each $j \in \{1, \dots, d\}$, $\gamma_j := \sum_{\mu} \boldsymbol{\alpha}_j^{\mu}$. So we also have that $\sum_j \gamma_j = 4m$.

Now we take the expectation with respect to $\mathcal{P}(u)$, and use the fact that the components are i.i.d. Rademacher so the result depends on the parity of the $\gamma_j$ in the following way:

$$\langle L, H_{\underline{\alpha}} \rangle = \begin{cases} \left( \frac{\eta^4 \kappa_4^g}{d^2} \right)^m & \text{if all } (\gamma_j)_{j=1,\dots,n} \text{ are even} \\ 0 & \text{if there is at least one } \gamma_j \text{ which is an odd number} \end{cases} \tag{100}$$

Hence, if we restrict to the set:

$$\tilde{\mathcal{I}}_m = \left\{ \underline{\alpha} \in \mathcal{I}_m \Big| \forall j \in \{1, \dots, d\} \sum_{\mu=1}^{n} \boldsymbol{\alpha}_j^{\mu} \text{ is even} \right\} \tag{101}$$

We have that all the indices belonging to $\tilde{\mathcal{I}}_m$ give the same contribution:

$$\langle L, H_{\underline{\alpha}} \rangle^2 = \left( \frac{\eta^4}{d^2} \kappa_4^g \right)^{2m} \tag{102}$$

Also, note that inside $\mathcal{I}_m$, $\boldsymbol{\alpha}! \leq (4!)^m$, so get the following estimates

$$\begin{aligned}
\|L_{n,d}^{\leq D}\|^2 &= \sum_{|\underline{\alpha}| \leq D} \frac{\langle L, H_{\underline{\alpha}} \rangle^2}{\underline{\alpha}!} \\
&\geq \sum_{m=0}^{\lfloor D/4 \rfloor} \sum_{\underline{\alpha} \in \tilde{\mathcal{I}}_m} \frac{\langle L, H_{\underline{\alpha}} \rangle^2}{\underline{\alpha}!} \\
&\geq \sum_{m=0}^{\lfloor D/4 \rfloor} \sum_{\underline{\alpha} \in \tilde{\mathcal{I}}_m} \left( \frac{\eta^4 \kappa_4^g}{\sqrt{4!} d^2} \right)^{2m} \\
&\geq \sum_{m=0}^{\lfloor D/4 \rfloor} \#\tilde{\mathcal{I}}_m \left( \frac{\eta^4 \kappa_4^g}{\sqrt{4!} d^2} \right)^{2m}
\end{aligned} \tag{103}$$

Now we just need to estimate the cardinality of $\tilde{\mathcal{I}}_m$. A lower bound can be provided by considering

$$\hat{\mathcal{I}}_m = \left\{ \underline{\alpha} \in \mathcal{I}_m \big| \forall \mu \in [n] \ \forall j \in [d] \ \underline{\alpha}_j^{\mu} \text{ is even} \right\} \tag{104}$$

Clearly $\hat{\mathcal{I}}_m \subseteq \tilde{\mathcal{I}}_m$ and $\#\hat{\mathcal{I}}_m = \binom{n}{m} \binom{d+1}{2}^m$ because first we can pick the $n - m$ rows of $\underline{\alpha}$ that will be $0 \in \mathbb{N}^d$, which can be done in $\binom{n}{m}$. Then, for each non-zero row, we need to pick two columns

(with repetitions) in which to place a 2 and leave all the other entries as 0, that can be done in $\binom{d+1}{2}^m$ ways.

So plugging the lower bound in the previous estimate, we reach the inequality that we wanted to prove:

$$||L_{n,d}^{\leq D}||^2 \geq \sum_{m=0}^{\lfloor D/4 \rfloor} \binom{n}{m}\binom{d+1}{2}^m \left(\frac{\eta^4 \kappa_4^g}{\sqrt{4!}d^2}\right)^{2m} = \sum_{m=0}^{\lfloor D/4 \rfloor} \binom{n}{m}\binom{d+1}{2}^m \left(\frac{\beta^2 \kappa_4^g}{\sqrt{4!}d^2(1+\beta)^2}\right)^{2m} \tag{105}$$

**Upper bound**  We start from (75) using the following rewriting:

$$||L_{n,d}^{\leq D}||^2 = \sum_{|\boldsymbol{\alpha}| \leq D} \frac{\langle L, H_{\boldsymbol{\alpha}}\rangle^2}{\boldsymbol{\alpha}!}$$

$$= \sum_{|\boldsymbol{\alpha}| \leq D} \frac{\mathbb{E}_{x \sim \mathbb{P}}\left[H_{\boldsymbol{\alpha}}(x)\right]^2}{\boldsymbol{\alpha}!} \tag{106}$$

$$= \sum_{|\boldsymbol{\alpha}| \leq D} \frac{1}{\boldsymbol{\alpha}!} \mathbb{E}_{u \sim \mathcal{P}(u)}\left[\prod_{\mu=1}^{n} \mathbb{E}_{x^\mu \sim \mathbb{P}(\cdot|u)}\left[H_{\boldsymbol{\alpha}^\mu}(x^\mu)\right]\right]^2$$

Now use 16 and plug in the formulas for the inner expectations.

$$||L_{n,d}^{\leq D}||^2 = \sum_{|\boldsymbol{\alpha}| \leq D} \frac{1}{\boldsymbol{\alpha}!} \mathbb{E}_{u \sim \mathcal{P}(u)}\left[\prod_{\mu=1}^{n} \frac{T_{|\boldsymbol{\alpha}^\mu|,g}}{d^{|\boldsymbol{\alpha}^\mu|/2}} u^{\boldsymbol{\alpha}^\mu}\right]^2$$

$$= \sum_{|\boldsymbol{\alpha}| \leq D} \frac{\left(\prod_{\mu=1}^{n} T_{|\boldsymbol{\alpha}^\mu|,g}\right)^2}{\boldsymbol{\alpha}! d^{|\boldsymbol{\alpha}|}} \mathbb{E}_{u \sim \mathcal{P}(u)}\left[u^{\boldsymbol{\alpha}}\right]^2 \tag{107}$$

Now we use our prior assumption that $u_i \overset{\text{i.i.d.}}{\sim} \text{Rad}(1/2)$. Note that odd moments of $u_i$ are equal to 0 and even moments are equal to 1, so (denoting by $\chi_A(\cdot)$ the indicator function of set $A$):

$$\mathbb{E}_{u \sim \mathcal{P}(u)}\left[u^{\boldsymbol{\alpha}}\right] = \prod_{i=1}^{d} \mathbb{E}_{u_i \sim \text{Rademacher}(1/2)}\left[u_i^{\sum_{\mu=1}^{n} \boldsymbol{\alpha}_i^\mu}\right] = \chi_{\left\{\sum_{\mu=1}^{n} \boldsymbol{\alpha}_i^\mu \text{ is even for all } i\right\}}(\boldsymbol{\alpha}) \tag{108}$$

Now the key point of the proof: this last formula, together with (92), implies that most of the addends in the sum in (107) are zero. Set $|\boldsymbol{\alpha}| = m$, then the set of multi-indices that could give a non-zero contribution is

$$\mathcal{A}_m = \left\{\boldsymbol{\alpha} \in \mathbb{N}^{n \times d} \,\big|\, |\boldsymbol{\alpha}| = m, \,\forall\, \mu \in [n], \boldsymbol{\alpha}^\mu = 0 \text{ or } |\boldsymbol{\alpha}^\mu| > 4, \text{ and } \forall\, i \in [d] \sum_{\mu=1}^{n} \boldsymbol{\alpha}_i^\mu \text{ is even}\right\} \tag{109}$$

Using this fact together with (82) we get:

$$||L_{n,d}^{\leq D}||^2 = \sum_{m=0}^{D} \sum_{\boldsymbol{\alpha} \in \mathcal{A}_m} \frac{\left(\prod_{\mu=1}^{n} T_{|\boldsymbol{\alpha}^\mu|,g}\right)^2}{\boldsymbol{\alpha}! d^{|\boldsymbol{\alpha}|}}$$

$$\leq \sum_{m=0}^{D} \sum_{\boldsymbol{\alpha} \in \mathcal{A}_m} \frac{1}{d^m} \prod_{\mu=1}^{n}\left(\left(\frac{\beta}{1+\beta}\right)^{|\boldsymbol{\alpha}^\mu|} \mathbb{E}\left[h_{|\boldsymbol{\alpha}^\mu|}(g)\right]^2\right) \tag{110}$$

$$\leq \sum_{m=0}^{D} \sum_{\boldsymbol{\alpha} \in \mathcal{A}_m} \frac{1}{d^m}\left(\frac{\beta}{1+\beta}\right)^m \sup_{k \leq m}\left(\mathbb{E}\left[h_k(g)\right]^{2m/k}\right)$$

$$= \sum_{m=0}^{D} \frac{1}{d^m}\left(\frac{\beta}{1+\beta}\right)^m \sup_{k \leq m}\left(\mathbb{E}\left[h_k(g)\right]^{2m/k}\right) \#\mathcal{A}_m$$

In the third step we used the following inequality

$$\prod_\mu \mathbb{E}\left[h_{|\boldsymbol{\alpha}^\mu|}(g)\right]^2 \leq \prod_\mu \sup_{k \leq m}\left(\mathbb{E}\left[h_k(g)\right]^{1/k}\right)^{2|\boldsymbol{\alpha}^\mu|} = \sup_{k \leq m}\mathbb{E}\left[h_k(g)\right]^{2m/k}$$

That holds since $\underline{\boldsymbol{\alpha}} \in \mathcal{A}_m$ hence $\sum_\mu |\boldsymbol{\alpha}^\mu| = m$.

It is hard to compute exactly the cardinality of $\mathcal{A}_m$, but we can estimate it by considering the inclusion $\mathcal{A}_m \subseteq \tilde{\mathcal{A}}_m$ defined as

$$\tilde{\mathcal{A}}_m = \left\{\boldsymbol{\alpha} \in \mathbb{N}^{n \times d} \big| |\underline{\boldsymbol{\alpha}}| = m, \text{ there are at most } \left\lfloor\frac{m}{4}\right\rfloor \text{ non-zero rows and } \left\lfloor\frac{m}{2}\right\rfloor \text{ non-zero columns}\right\} \tag{111}$$

Assume now $m > 0$ (the case $m = 0$ can be treated separately, leading to the addend +1 in (74)). To compute the cardinality of $\tilde{\mathcal{A}}_m$ we just have to multiply the different contributions

- There are $\binom{n}{\lfloor m/4\rfloor}$ possibilities for the non-zero rows

- There are $\binom{d}{\lfloor m/2\rfloor}$ possibilities for the non-zero columns

- once restricted to a $\lfloor m/4\rfloor \times \lfloor m/2\rfloor$ we have to place the units to get to norm $m$. It is the counting problem of placing $m$ units inside the $\lfloor m/4\rfloor\lfloor m/2\rfloor$ matrix entries. The possibilities are

$$\binom{\lfloor m/4\rfloor\lfloor m/2\rfloor + m - 1}{m} \tag{112}$$

So we get the following estimate for the cardinality of $\mathcal{A}_m$

$$\#\mathcal{A}_m \leq \binom{n}{\lfloor m/4\rfloor}\binom{d}{\lfloor m/2\rfloor}\binom{\lfloor m/4\rfloor\lfloor m/2\rfloor + m - 1}{m} \tag{113}$$

Plugging into (110) we reach the final formula (74), which completes the proof.

**Proof of corollary 15**    Now we turn to the proof of 15 on the asymptotic behavior of the bound

$$\binom{n}{m} = \frac{n^m}{m!} + O(n^{m-1}) \tag{114}$$

$$\binom{d+1}{2}^m \geq \frac{d^{2m}}{2^m} \tag{115}$$

So we have that:

$$||L_{n,d}^{\leq D(n)}||^2 \geq \sum_{m=0}^{\lfloor D/4\rfloor}\left(\frac{n^m}{m!} + O(n^{m-1})\right)\left(\frac{d^{2m}}{2^m} + O(1)\right)\left(\frac{\beta^2 \kappa_4^g}{\sqrt{4!}d^2(1+\beta)^2}\right)^{2m}$$

$$= \left(\sum_{m=0}^{\lfloor D/4\rfloor}\frac{1}{m!}\left(\frac{\beta^2\kappa_4^g}{\sqrt{2}\sqrt{4!}(1+\beta)^2}\right)^{2m}\left(\frac{n}{d^2}\right)^m\right) + O\left(\sum_m \frac{n^{m-1}}{m!d^{2m}}\left(\frac{\beta^2\kappa_4^g}{\sqrt{2}\sqrt{4!}(1+\beta)^2}\right)^{2m}\right) \tag{116}$$

Then we lower-bound the sum by considering just the last term.

$$||L_{n,d}^{\leq D(n)}||^2 \geq \frac{1}{\lfloor D/4\rfloor!}\left(\frac{\beta^2\kappa_4^g}{\sqrt{2}\sqrt{4!}(1+\beta)^2}\right)^{2\lfloor D/4\rfloor}\left(\frac{n}{d^2}\right)^{\lfloor D/4\rfloor} + o\left(\frac{1}{\lfloor D/4\rfloor!}\left(\frac{Cn}{d^2}\right)^{\lfloor D/4\rfloor}\right) \tag{117}$$

So by using $k! \leq k^k$ on $\lfloor D/4\rfloor$ and picking $n, d$ large enough so that numerical constants and the $o(\dots)$ become negligible for the estimate, we get (10).

Plugging in the scaling $n \asymp d^\theta$ and $D(n) \asymp \log^{1+\varepsilon}(n)$ it is clear that what decides the behaviour of the sequence is the term $\frac{n}{d^2}$.

- If $0 < \theta \leq 2$, $\frac{n}{d^2} \to 0$ and so (10) does not provide information on the divergence of the LDLR norm.

- If $\theta > 2$ it is $\frac{n}{d^2} \to \infty$ faster that the logarithmic term at denominator, so the right hand side in (10) diverges at $\infty$, proving the second regime in (12)

Now let us turn to proving (11). We are in the regime $m \leq D << \min(n, d)$, hence we can use the following estimates of binomial coefficients and factorial

$$\binom{n}{\lfloor m/4 \rfloor} \leq n^{m/4}$$
$$\binom{d}{\lfloor m/2 \rfloor} \leq d^{m/2}$$
$$\binom{\lfloor m/4 \rfloor \lfloor m/2 \rfloor + m - 1}{m} \leq \left(m^2/8\right)^m \tag{118}$$
$$m! \leq m^m$$

on 74 to get

$$||L_{n,d}^{\leq D}||^2 \leq 1 + \sum_{m=1}^{D} \left(\frac{\beta}{1+\beta}\right)^m m^{2m} \sup_{k \leq m} \left(\mathbb{E}\left[h_k(g)\right]^{2m/k}\right) \left(\frac{n}{d^2}\right)^{m/4} \tag{119}$$

Now we use estimate the term depending on the Hermite coefficients of $g$, using the assumption 5:

$$\sup_{k \leq m} \left(\mathbb{E}\left[h_k(g)\right]^{2m/k}\right) \leq \sup_{k \leq m} \left((\Lambda^k k!)^{2m/k}\right)$$
$$\leq \Lambda^{2m} \sup_{k \leq m} \left(k^{2m}\right) \tag{120}$$
$$\leq \Lambda^{2m} m^{2m}$$

Plugging into the estimate for the LDLR we get (11)

$$||L_{n,d}^{\leq D}||^2 \leq 1 + \sum_{m=1}^{D} \left(\frac{\Lambda^2 \beta}{1+\beta}\right)^m m^{4m} \left(\frac{n}{d^2}\right)^{m/4} \tag{121}$$

By letting $n \asymp d^\theta$, for $\theta > 0$ and $D(n) \asymp \log^{1+\varepsilon}(n)$, we can see that the bound goes to 1 when $\theta < 2$, proving the first regime in (12).

### B.7 Limitations of our theoretical analysis

The main limitation of the theoretical portion of our work is that it relies on the assumption that the null hypothesis is standard i.i.d. Gaussian noise. Since this assumption is central to the large body of work analysing hypothesis tests [37], a very interesting future direction is to develop tools for analysing the case of a different null hypothesis. On the technical side, while the assumption of Rademacher prior on $u$ is not essential and could be easily generalized to other isotropic distributions, assumption 1 is a key requirement to carry out the proofs. We discuss the limitations of the random feature analysis, and in particular the Gaussian equivalence theorem [79–83], at the end of section 4.1.

As is generally the case in high-dimensional statistics, our results do not constitute a complete proof of the existence of a statistical-to-computational gap: our argument relies on the low-degree conjecture 4, and the divergence of the norm of the likelihood ratio is only a necessary condition for *strong distinguishability*. In fact, there are no techniques at the moment that can prove that average-case problems require super-polynomial time in the hard phase, even if we assume $P \neq NP$ [37]. So our work should be seen as providing rigorous evidence for a statistical-to-computational gap in the spiked cumulant model.

## C  Details on the replica analysis

We analytically compute the generalisation error of a random feature model trained on the Gaussian mixture classification task of B.2.3 by exploiting the Gaussian equivalence theorem for dealing with structured data distributions [79–83]. In particular, we use equations derived by Loureiro *et al.* [78]

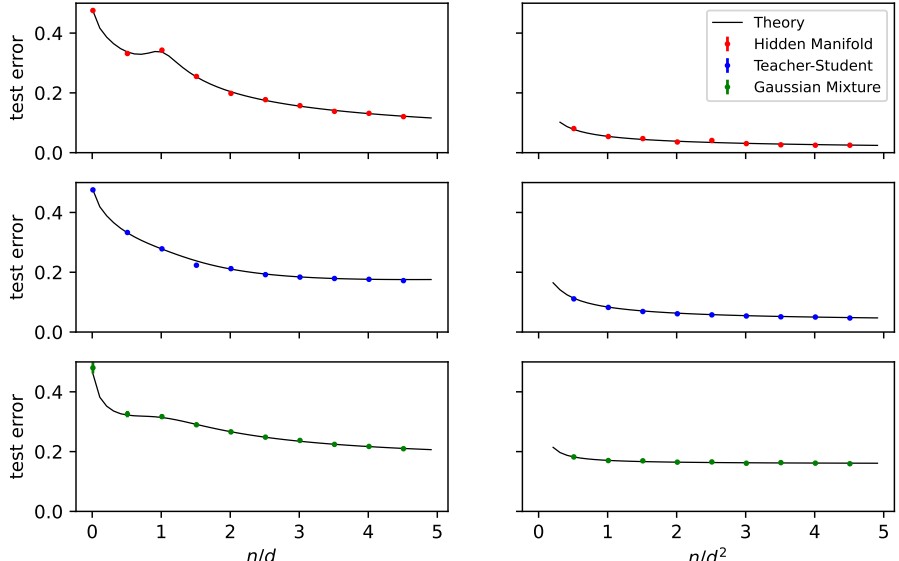

Figure 8: **Linear and quadratic sample regimes for different synthetic data models.** (*Right*) Generalization error of the hidden manifold model (top), the teacher-student setup (center) and a mixture of two Gaussians as a function of the ratio of the number of samples and the input dimension. (*Left*) Same except that the number of samples scales with the square of the input dimension. The solid black line corresponds to the replica theory prediction while the coloured dots display the outcome of the numerical simulations averaged over 10 different seeds. In all the panels, $d = 1000$ and $d = 20$ for linear and quadratic sample regimes respectively, $\lambda = 0.01$ for both the teacher–student setup and the hidden manifold model while $\lambda = 0.1$ for Gaussian mixtures. In the Gaussian mixture case, $\mu_\pm = (\pm 1, 0, ..., 0) \in \mathbb{R}^d$ while the covariance matrices are both isotropic and, in particular, both equal to the identity matrix: $\Sigma_\pm = \mathbb{I}$.

that describe the generalisation error of random features on Gaussian mixture classification using the replica method of statistical physics. The equations describe the high-dimensional limit where $n, d$, and the size of the hidden layer $p \to \infty$ while their ratios stay finite, $\alpha = n/d$ and $\gamma = d/p \sim O(1)$. In this regime, the generalisation error is a function of only scalar quantities, i.e.

$$\epsilon_g = 1 - \rho_+ \mathbb{E}_\xi \left[ \text{sign}\left(m_+^\star + \sqrt{q_+^\star}\xi + b^\star\right) - \right] + \rho_- \mathbb{E}_\xi \left[ \text{sign}\left(m_-^\star + \sqrt{q_-^\star}\xi + b^\star\right)\right], \qquad (122)$$

where $\xi \sim \mathcal{N}(0,1)$, $\rho_\pm$ defines the fraction of data points belonging to class $y = \pm 1$ while $m_\pm^\star$, $q_\pm^\star$ and $b^\star$ are the so-called overlap parameters and bias term respectively. Their value for every size of the training set can be determined by solving the following optimisation problem [78]:

$$f_\beta = \underset{\{q_k, m_k, V_k, \hat{q}_k, \hat{m}_k, \hat{V}_k, b\}_{k=+,-}}{\text{extr}} \left[ \sum_{k=+,-} -\frac{1}{2}\left(\hat{q}_k V_k - q_k \hat{V}_k\right) + m_k \hat{m}_k + \lim_{p\to\infty} \frac{1}{p}\Psi_s + \alpha\gamma\Psi_e \right]. \tag{123}$$

where the entropic potential $\Psi_s$ can be expressed as a function of the means $\mu_\pm \in \mathbb{R}^p$ and covariances $\Sigma_\pm \in \mathbb{R}^{p\times p}$ of the hidden-layer activations of the random features model, while the energetic potential $\Psi_e$ depends on the specific choice of the loss function $\ell(\cdot)$.

Solving the optimization problem in 123, leads to a set of coupled saddle-point equations for the overlap parameters and the bias term. This set of equations is nothing but a special case of the equations already derived in [78], that is high-dimensional classification of a mixture of two Gaussians with random features, except for the typo in the self-consistency equation of the bias term

$$b = \left( \sum_{k=\pm} \frac{\rho_k}{V_k} \right)^{-1} \sum_{k=\pm} \rho_k \mathbb{E}_\xi \left[ \eta_k - m_k \right], \tag{124}$$

with $\eta_k$ being the extremizer

$$\eta_k = \operatorname*{argmin}_\lambda \left[ \frac{\left( \lambda - \sqrt{q}_k \xi - m_k - b \right)^2}{2V_+} + \ell \left( y_k, \lambda \right) \right]. \tag{125}$$

and $\ell \left( \cdot \right)$ being any convex loss function. Analogously to the more general setting of [78], the Gaussian Equivalence Theorem allows to express the entropic potential $\Psi_s$ as a function of the means $\boldsymbol{\mu}_k \in \mathbb{R}^p$ and covariances $\Sigma_k \in \mathbb{R}^{p \times p}$ of the random features

$$\Psi_s = \lim_{p \to \infty} \frac{1}{p} (\hat{q}_+ \boldsymbol{\mu}_+ + \hat{q}_- \boldsymbol{\mu}_-)^t \left( \lambda \mathbb{I}_p + \hat{V}_+ \Sigma_+ + \hat{V}_- \Sigma_- \right)^{-1} (\hat{q}_+ \boldsymbol{\mu}_+ + \hat{q}_- \boldsymbol{\mu}_-) +$$
$$+ \lim_{p \to \infty} \frac{1}{p} \operatorname{Tr} \left( (\hat{q}_+ \Sigma_+ + \hat{q}_- \Sigma_-) \left( \lambda \mathbb{I}_p + \hat{V}_+ \Sigma_+ + \hat{V}_- \Sigma_- \right)^{-1} \right); \tag{126}$$

while the energetic potential $\Psi_e$ depends on the specific choice of the loss function $\ell \left( \cdot \right)$

$$\Psi_e = -\frac{1}{2} \mathbb{E}_\xi \left[ \sum_{k=\pm} \rho_k \frac{\left( \eta_k - \sqrt{q}_k \xi - m_k - b \right)^2}{2V_k} + \ell \left( y_k, \eta_k \right) \right]. \tag{127}$$

As discussed in 4.1, the Gaussian Equivalence Theorem breaks in the quadratic sample regime if the data are sampled from the spiked Wishart model. Interestingly, this is not the case for the Hidden Manifold model. Indeed, as shown in fig. 8, we still get a perfect match between the Gaussian theory and the numerical simulations even in the quadratic sample regime. The theoretical prediction can be easily derived by rescaling the free-energy associated to the Hidden Manifold model as in [80] by a factor $1/d^2$. This is the same trick already proposed in [84] for teacher–student settings and displayed in the middle panel of fig. 8.

