# OpenReview forum: "Learning from higher-order correlations, efficiently: hypothesis tests, random features, and neural networks"
_NeurIPS.cc/2024/Conference — NeurIPS 2024 poster_

### Official Review · Reviewer_GzyK · 2024-07-11

**Soundness:** 3
**Presentation:** 3
**Contribution:** 3
**Rating:** 5
**Confidence:** 4

**Summary:**

The authors calculate the likelihood ratio (LR) norm (=E_Q[L^2]), of the LR L, for a class of non-gaussian distributions. In particular they show in their Theorem 2 that LR norm  diverges as n=d^c for c>1, where d is the dimension and n is the sample size. Note that LR diverges when the distributions are distinguishable in a strong sense. Similarly in Theorem 5 the authors define a low degree LR as the the projection of L to the space of degree D polynomials, and again show that it diverges as n=d^c for c>1.

**Strengths:**

The results are interesting and theorems are well written.

**Weaknesses:**

While the results are interesting and novel, the proofs follow relatively easily from elementary techniques. So from a purely theoretical perspective, I am not certain of how important the contribution will be.

**Questions:**

- The expressions of the two LR norms are too complicated to gain any insight. If you are working with large n and d then can't they be simplified further?
- Presumably the KL divergence is much harder to calculate than the LR norm?

**Limitations:**

No societal impact concerns, theory paper

---

> ### Author Rebuttal · Authors · 2024-08-07
>
> We thank you for your comments. We hope that our response can answer your questions; if so, we would appreciate it if you could revisit the rating of
> the paper. If not, we are happy to give further explanations during the
> discussion period.
>
> First we would like to address a possible source of confusion, likely due to a
> typo:
>
> > Similarly in Theorem 5 the authors define a low degree LR as the the
> > projection of L to the space of degree D polynomials, and again show that it
> > diverges as n=d^c for c>1.
>
> Theorem 5 proves that the number of samples required for computational
> distinguishability, i.e. for distinguishing the two distributions in polynomial
> time, is $n\asymp d^c$ with $c>2$, whereas for statistical distinguishability it
> is $c>1$. Hence there is a large *statistical-to-computational* gap, described in section 3.3.
>
> > While the results are interesting and novel, the proofs follow relatively
> > easily from elementary techniques. So from a purely theoretical perspective, I
> > am not certain of how important the contribution will be.
>
> We appreciate that our results are considered interesting and novel. We spent a
> lot of time refining and simplifying the proofs and our writing so as to make
> the article as accessible as possible, so some technical difficulties may not be
> apparent from the final forms. In particular we want to highlight that...
>
> - ... we made a specific effort to find the minimum set of assumptions that allowed
>   us to keep the distribution of the latent variable $g$ general;
> - ... lemma 13 required careful use of assumption 1 to bound the complicated
>   functions $f$ and then the use of combinatorial-probabilistic identities to
>   conclude the proof.
>
> In light of the large, recent interest in analysing computational-to-statistical
> gaps in the Gaussian additive model (cf. refs. 9, 14-24, 25-33, many of which
> appeared at NeurIPS and related conferences), we are confident that our precise
> analysis of a **non-Gaussian** data model will be of great interest to the
> NeurIPS community, which deals with machine learning problems where
> **higher-order, non-Gaussian** data correlations are key (cf. our
> introduction). In particular, we believe that providing _exact_ formulas
> for the LR, rather than just estimates, will be a useful starting point for
> future applications of the spiked cumulant model, and that the algorithmic
> results on the low-degree likelihood ratio will inspire further work in this
> direction.
>
> > The expressions of the two LR norms are too complicated to gain any
> > insight. If you are working with large n and d then can't they be simplified
> > further?
>
> We did not find obvious ways to simplify formulas (5) and (6) further since the
> precise value of $f$ depends strongly on the distribution of the non-Gaussianity
> $g$. It is however possible to gain insights from Eqs (5) and (6) in two ways:
> by studying $f$ and then performing bounds; or by making additional assumptions
> on the non-Gaussianity $g$.
>
> The first method is how we extracted the presence of a phase transition at
> $\theta = 1$ for strong statistical distinguishability from the formula, as
> discussed below Eq (6); we will expand this discussion with additional details
> from appendix B.5.3 should the paper be accepted. On one hand the lower bound
> $||L_{n,d}||^2 \ge \frac{\max_\lambda f(\beta,\lambda)^n}{2^d}$, implies the divergence of the LR as soon as $n\ge d$ and $f$ is not
> constantly 1. On the other hand lemma 13 uses the sub-Gaussianity of $g$ to deduce $||L_{n,d}||^2 \lesssim \sum_{k=0}^n C
> \left(\frac{C_{\beta,g}n}{d}\right)^k$, which leads to the upper bound on the LR for $\theta<1$.
>
> By choosing an explicit distribution for $g$ (the second method), we were able
> to carry the computations even further, as we did at the end of B.5.3 for the
> $g\sim$ Rademacher(1/2) case (see also Figures 4 and 5 where we are able to use formulas (5) and (6) to plot exactly the growth of LR norm). This allows us to delineate the precise thresholds for the effective sample complexity $\gamma$ and the signal-to-noise ratio $\beta$ of a phase transition in the linear regime for $n \asymp \gamma d$.
>
> > Presumably the KL divergence is much harder to calculate than the LR norm?
>
> This is an interesting question. By Jensen's inequality, the LR always provides a bound on the KL divergence:
> $$ D_{KL}( ℙ ||
> ℚ )=\underset{  x\sim ℙ } E [\log L_{n,d}(  x)]\le \log
> \underset{  x\sim ℙ } E [L(x)]=\log
> ||L_{n,d}||^2$$
> However, computing an explicit expression for the KL divergence is more complicated, as the referee suspected, since the logarithm creates difficulties in exchanging integrals over the inputs $x$ with the integrals over the spike $u$ and the latent variable $g$ as we do in the proof of theorem 2. However, in the case of the spiked cumulant model we can again use Jensen's inequality on the inner expectations in formula (61) to obtain
> $$D_{KL}(ℙ || ℚ )\ge
> \underset{ (x^1,\dots,x^n)\sim ℙ } {E} \left[\underset{u} E \left[\log\left(\prod_{\mu=1}^nl(x^\mu|u)\right)\right]\right]=n\underset{u}E \left[\underset{x\sim ℙ } {E }\left[\log\left(l(x|u)\right)\right]\right]$$
> which could lead to an interesting lower bound either by a direct computation of $\log\left(l(x^\mu|u)\right)$ for an explicit choice of $g$, or by a further application of Jensen inequality to pass the log inside the expectation over $g$ in (59). In the end, we focused on the LR because it is the optimal statistical test in the sense of
> trading off type I and type II statistical errors (falsely rejecting vs. falsely
> _failing_ to reject the null hypothesis), but we will add this discussion to the revised version of the paper.

---

### Official Review · Reviewer_xzwG · 2024-07-12

**Soundness:** 4
**Presentation:** 3
**Contribution:** 4
**Rating:** 7
**Confidence:** 2

**Summary:**

The paper develops a likelihood ratio (LR) for a LR test to distinguish between two distributions, a "spiked" model and a Gaussian one related to higher order cumulants. Using the LR, the authors prove that in an ideal scenario the minimum number of samples needed to distinguish the two distributions is linear whereas in a more practical computational scenario that number is quadratic. Furthermore, they show that neural networks can effectively learn to distinguish this classification task whereas random feature models (i.e. kernel learners) are not able to due to a mismatch in approximation power compared to target function complexity. Thus, feature learner based models outperform fixed feature map models for this particular task.

**Strengths:**

The paper builds on top of existing literature and provides a significantly original statistical insight for an existing problem. This is displayed in a clear and well written manner. This is a significant contribution to what is a difficult task as it provides statistical guarantees regarding sample sizes in both theory and practical regimes.

**Weaknesses:**

I found a typo:

Eq (1) - I believe it should be $\tilde{x}^\mu$ as it is later referred to in line 132.

**Questions:**

None that I can think of.

**Limitations:**

Everything is adequately addressed.

---

> ### Author Rebuttal · Authors · 2024-08-07
>
> We thank you for the very positive feedback, and for carefully reading the manuscript. Your summary perfectly captures the main ideas of our paper, and we thank you for pointing out the typo.

---

> > ### Comment · Reviewer_xzwG · 2024-08-12
> >
> > Having given a re-read of the paper, reading the other reviews, and your rebuttals, I still stand by the score I listed; however, with an updated confidence. I believe that the paper tackles a specific, statistically technical problem and does so in a thorough manner. I do think that the questions raised by reviewer u8Z4 and the subsequent clarification by the authors would help motivate the experimental sections in the paper better.

---

### Official Review · Reviewer_mvai · 2024-07-13

**Soundness:** 3
**Presentation:** 3
**Contribution:** 3
**Rating:** 6
**Confidence:** 3

**Summary:**

The paper provides an in-depth examination of the efficiency of neural networks in extracting features from higher-order cumulants in high-dimensional datasets. Higher-order cumulants, which quantify non-Gaussian correlations among multiple variables, are crucial for neural network performance. The study focuses on the spiked cumulant model to determine the statistical and computational limits of recovering specific directions from these cumulants. Key findings include the following: (i)The number of samples $n$ required to distinguish between spiked cumulant inputs and isotropic Gaussian inputs. (ii) An exact formula for the likelihood ratio norm, indicating that statistical distinguishability requires $n \approx d $ samples, whereas polynomial time algorithms require $ n \approx d^2 $ samples. (iii) Numerical experiments showing that neural networks can efficiently learn to distinguish between distributions with quadratic sample complexity, while "lazy" methods like random features perform no better than random guessing.

**Strengths:**

1)	The paper provides a thorough theoretical foundation by examining the statistical and computational limits of learning from higher-order cumulants (HOCs), demonstrating that unbounded computational power requires a number of samples linear in the input dimension for statistical distinguishability.
2)	The study offers new insights into the quadratic sample complexity of learning from HOCs for a wide class of polynomial-time algorithms and validates these theoretical findings with numerical experiments, highlighting the efficiency of neural networks compared to random features.
3)	By using fundamental limits on learning from HOCs to benchmark neural networks, the research shows a clear separation in learning efficiency between neural networks and random features, and extends its relevance to practical models for images, showcasing the crossover behavior of higher-order cumulants in real-world applications.

**Weaknesses:**

1)	The study assumes a Rademacher prior for the special direction $u$. While not essential, this assumption could restrict the generalizability of the results to other isotropic distributions. A detailed discussion on this assumption would be useful.
2)	The paper primarily uses the second moment method for distinguishability, which might not fully capture the variety of hypothesis tests that could be relevant for higher-order cumulants.

**Questions:**

1)	It is stated that theorem 2 implies the presence of a phase transition at $\theta=1$ for the strong statistical distinguishability. Can authors provide further insight on the consequences of theorem 2 with numerical results around the phase transition point?
2)	For the computational distinguishability, theorem 5 identifies an hard phase ($1<\theta<2$) from an easy phase ($\theta>2$). Is this result restricted to the underlying single spike model? If so, how it can be generalized to a generic covariance model?
3)	How does the spiked cumulant model differ from the spiked Wishart model in terms of classification difficulty?
4)	How does whitening the inputs affect the increased difficulty in classification as described in theorem 5?
5)	What would happen if the inputs were not whitened, particularly regarding the degree-2 polynomials and their contributions to the LDLR ?
6)	How does the sample complexity differ with and without whitening in the spiked cumulant model compared to the spiked Wishart model?
7)	The likelihood ratio norm with 𝑔∼Rademacher(1/2) diverges for 1<θ<2, indicating that the likelihood ratio test can distinguish between the hypotheses statistically. However, this divergence also highlights the computational complexity, as achieving this distinguishability may not be feasible in polynomial time. If so, what is the main reason for assuming the distribution for $g$ as such?

**Limitations:**

No potential negative impact of their work has been identified.

---

> ### Author Rebuttal · Authors · 2024-08-07
>
> We thank you for your useful comments and questions that will
> undoubtedly help to improve the paper. We will now address all the points one by
> one.
>
> > Assumption of a Rademacher prior for the special direction u.
>
> Following your suggestion, we will move the discussion of the prior on $u$ that
> is currently in appendix B.4.1 to the main text, should the paper be
> accepted. We make the Rademacher assumption to make the derivation of the
> results less heavy, but we fully expect our results to also hold for other
> standard choices like having $u$ distributed uniformly on the sphere or with
> independent i.i.d Gaussian components. An idea on how to adapt the proofs can be
> seen in Theorem 8 in appendix B.2.3.
>
> > Second moment method for distinguishability.
>
> We agree that analysing different measures of distinguishability, like the
> Kullback-Leibler divergence, is an interesting direction for future work. Here,
> we focused on the second moment method for the likelihood ratio since (1) it is
> the optimal statistical test in the sense of trading off type I and type II
> statistical errors (falsely rejecting vs. falsely _failing_ to reject the null
> hypothesis) and because (2) it has been used to derive statistical thresholds
> for many high-dimensional inference problems, cf. [22],[27],[37],[94].
>
> > Can authors provide
> > further insight on the consequences of theorem 2 with numerical results around
> > the phase transition point?
>
> We thank the reviewer for this suggestion. In Fig. 1 of the reply figures
> submitted above, we show a phase transition in the success rate of a
> spike-recovery algorithm based on exhaustive search right around $\theta=1$. The
> algorithm simply performs a maximum log-likelihood test along $v\cdot
> x^\mu$ for _all_ the possible spikes in the $d$-dimensional hypercube using formula
> (59) and outputs the most likely $v$. Even though we ran the algorithm only up
> to dimension 25 due to its exponential complexity (in $d$), the success rate of
> retrieving the correct spike shows strong hints of a phase transition precisely
> around the value of $\theta=1$ that is predicted by our theory.
>
> > Theorem 5 identifies an hard phase
> > ($1<\theta<2$) from an easy phase ($\theta>2$). Is this result restricted to the
> > underlying single spike model? If so, how it can be generalized to a generic
> > covariance model?
>
> Yes, the result applies precisely to the single spike model introduced in
> section 2. There are two ways to generalize to a different covariance matrix:
>
> 1. By changing only the covariance of the alternative distribution, the problem
>    becomes easier and the sample complexity exponent for computational
>    distinguishability would go back to 1.
> 2. Having different covariance matrices in both classes would correspond to
>    non-isotropic priors on $u$ and constitutes an interesting and complex future
>    direction. A rigorous analysis would present novel challenges that depend on
>    the details of the covariance matrix chosen.
>
> > Differences between spiked cumulant model and spiked Wishart model in
> > terms of classification difficulty?
>
> The spiked cumulant task is more difficult to learn:
>
> - Random feature (RF) models cannot learn it with linear or quadratic sample
>   complexity regions, while they do learn the spiked Wishart model;
> - Fully-trained two-layer networks learn the spiked cumulant task only with
>   quadratic sample complexity, while they learn the spiked Wishart task with
>   linear sample complexity already.
> - Finally, the spiked Wishart model does not present a
>   statistical-to-computational gap as the spiked cumulant model. Mathematically speaking, statistical distinguishability in the spiked Wishart model is impossible if $n\lesssim \frac{d}{\beta}$ , while for $n\gtrsim \frac{d}{\beta}$ spectral methods can provide, computationally fast, strong distinguishability (see [9-10]).
>
> > [Questions on whitening] 1. Whitening and difficulty in classification described in theorem 5 2. Whitening and degree 2 polynomials 3.Changes in sample complexity with and without withening
>
> If the inputs were not whitened, the leading eigenvector of the (empirical)
> covariance matrix (which is a degree-2 polynomial) would already be correlated
> with the spike, and hence simple PCA would be enough to partially recover the
> spike, yielding the sample complexities as in the spiked Wishart model. Hence
> there would be no need to extract information from the higher-order
> cumulants. Since we are interested in the ability of neural networks to extract
> information from non-Gaussian fluctuations, i.e. higher-order cumulants of order
> three and higher, we whiten the inputs to remove the spike from the covariance
> and force any algorithm therefore to extract the spike from the higher-order
> cumulants. The spiked cumulant problem therefore requires $n \gtrsim d^2$
> samples, making it harder than the spiked Wishart model with requires only $n
> \gtrsim d$.
>
> > Reasons for assuming the distribution for $g$ to be Rademacher$(1/2)$
>
> The choice of $g\sim$ Rademacher$(1/2)$ was mainly to provide a concrete
> application, since it leads to the problem of distinguishing a standard
> Gaussian from a bimodal Gaussian mixure with same mean and covariance matrix. We
> stress that Theorem 2 works for a wide range of distributions for $g$ (see
> appendix B.4.2), and it is sufficient to compute $f$ on the preferred
> distribution to find other examples. We expect them to share behaviour for
> $\theta>1$.
>
> ## Summary
>
> We want to thank you again for your careful reading of the paper
> and your questions. We hope that we have clarified your questions; if so, we
> would appreciate it if you increased the rating of the paper. If not, we're
> looking forward to clarifying any remaining doubts during the discussion.

---

> ### Comment · Reviewer_mvai · 2024-08-13
>
> Thank you for the detailed response and the additional results for the rebuttal. You have also clarified most of my raised questions and I will keep my initial score.

---

### Official Review · Reviewer_u8Z4 · 2024-07-14

**Soundness:** 3
**Presentation:** 3
**Contribution:** 4
**Rating:** 6
**Confidence:** 4

**Summary:**

The authors study the computational-to-statistical gap when learning from higher-order cumulants. In particular, they consider binary classification of the “spike cumulant model” introduced in [38]: inputs in both classes are drawn from a standard Gaussian distribution and are distinguished only by the addition of a spike to a higher-order cumulant in the distribution of one class. First, they provide bounds on the sample complexity necessary for statistical distinguishability: $n \gtrsim d$, and $n \gtrsim d^2$ for polynomial-time algorithms. Then, they perform numerical simulations showing that one-hidden-layer networks can learn both the spike cumulant model and a simple non-Gaussian model for images with $\sim d^2$ samples while random features cannot.

**Strengths:**

Overall the paper is well-written and the derivations are presented in a clear way. I have really appreciated the pedagogical structure of appendices A and B. To the best of my knowledge, the relevant literature has been appropriately covered.

Understanding how neural networks can extract information from higher-order-correlations in the data is an intriguing and relevant question that can shed light on open puzzles in theoretical machine learning. The spiked cumulant model is a paradigmatic setting that is well suited to address these questions from the theory standpoint, and theorem (2) with the associated phase transition is an interesting contribution.

**Weaknesses:**

The experimental section is somewhat underdeveloped, particularly considering it is presented as a central contribution of the paper. The existence of a gap between neural networks and random features in an interesting observation, however simulations on random features do not seem necessary given that their inability to learn can be inferred directly because the target function is beyond their approximation power. On the other hand, experiments for neural networks are limited to a very specific setting and narrow choice of parameters.

The purpose of the replica analysis in Sec. is not clear, especially because the curves are shown only for RFs in the linear regime (Fig. 1a, 2a) where we know RFs cannot learn. Moreover, the comparison between replicas and early stopping error is not strictly justified given that replicas describe the asymptotic performance. Several arbitrary choices impact this observation, such as the number of epochs = 200 and the regularization.

The purpose of Sec. 4.1 is not clear, given that the main focus of the paper is learning HOCs. This section could be moved to the appendix to expand Sec.s 4.2, 4.3.

Minor comments:

- Line 166: a square is missing in the LR norm definition
- Some choices of hyper-parameters appear arbitrary and are not motivated, e.g., m=5d, learning rates, epochs, regularization.
- The symbol $\alpha$ is used to indicate different quantities, e.g., eq. (11) and (120)

**Questions:**

How much is the recovery of the spike for NNs at $n \gtrsim d^2$ affected by the initial macroscopic overlap in Fig. 2? It would be useful to have a clarification on the role of finite size effects in this problem.

**Limitations:**

The authors have adequately addressed the limitations of the theoretical derivations in the paper.

---

> ### Author Rebuttal · Authors · 2024-08-07
>
> We thank you for your kind comments and the useful suggestions that
> will certainly improve the quality of the paper, and in particular our
> presentation of our experimental results. We reply to each suggestion and
> question in the following.
>
> ### Experimental setting
>
> In a nutshell, the idea behind our experimental section is to first _validate_
> our algorithms -- random features (RF) and neural networks (NN) -- on the
> simpler spiked Wishart task, before applying them to the spiked cumulant model,
> which is statistically very close to spiked Wishart, except of course for the importance of HOCs.
>
> - **Validating NN on spiked Wishart** In particular, for NN on spiked Wishart, we optimised all relevant
> hyper-parameters (network width, learning rate, weight decay) and quickly found
> a combination that worked well for both linear and quadratic sample complexity
> (for clarity, we only show one setting). Note that we are only interested
> whether weak recovery, i.e. better-than-chance performance, is possible. Hence
> as soon as we find a set of hyper-parameters where an algorithm reliably
> succeeds in finding the spike, we're done.
>
> - **Validating RF on spiked Wishart** For random features on the spiked Wishart, we found decent performance in the
> quadratic sample complexity regime, but not for linear sample complexity. This
> could be expected by the intuition behind some expressibility results, but to the best of our knowledge,
> these results are confined to isotropic Gaussian inputs and hence do not apply to spiked models [see e.g. Misiakiewicz &
> Montanari: Six Lectures on Linearized Neural Networks. arXiv:2308.13431]. Furthermore, it's
> not a priori clear whether the poor performance is due to the difficulty of the
> task, or to finite-size effects. Hence
> we turned to a replica analysis, which provides a sharp analysis down to the
> constants and was able to resolve the finite-size effects in this case.
>
> - **Testing RF and NN on spiked cumulant** Having thus validated the performance of NN and RF, we can turn to the spiked
> cumulant model. Note that the only difference to the previous experiments is
> (1) we now have whitened the data, so the covariance is uninformative; and (2) that we change the distribution of the latent variable $g$ from Gaussian to Rademacher; these distributions have the same mean and covariance. Neural networks require the
> quadratic sample complexity to solve the task as predicted by our theory, while random features
> fail. Since we only changed the statistics of the latent variable, this highlights the data-driven separation of the two methods. As we discuss, the replica analysis
> does not extend to this case, making the validation on the spiked Wishart model
> all the more important. Note that we didn't find neural networks learning the
> spiked cumulant with linear sample complexity in any combination of
> hyper-parameters that we tried, even if we report only one set for clarity.
>
> - **Comparing early-stopping performance vs. replicas**: Since we ran the networks long enough for
> convergence and optimised the hyperparameters, we expect the early stopping
> error to be very close to their asymptotic behaviour, so the replicas should be a
> good comparision. We see that the errors of replicas and the random
> features overlap (where they start to learn due to finite size effects); so our
> experiments show that the assumption of the random features being close to the
> asymptotic behaviour is true and thus our comparision is justified.
>
> We now appreciate that this line of thought was not clear enough in the
> manuscript, and will add some comments in the revised version, should the paper
> be accepted.
>
> ### Question on initial overlap in simulations
>
> The initial macroscopic overlap in the simulations with neural networks arises
> from the fact that we plot the maximum overlap over many (order of the input
> dimension $d$) hidden neurons, hence even for random initial weights, a few
> neurons will have large overlaps, but they do not affect our results: in
> response to your question, we reran the simulations for the fully-trained
> networks, but manually ensured that the initial weight of each neuron has only a
> $1 / \sqrt d$ overlap with the spike. We show the results in the rebuttal
> figure 2, which indeed do not change much from the completely random
> initialisation. Our hypothesis is that the dynamics of the wide network is
> dominated by the majority of neurons, which do not have a macroscopic overlap.
>
> ### Minor comments
>
> Thank you for checking for typos and consistent notation, we have corrected the
> manuscript and made the use of $\alpha$ consistent.
>
> ### Summary
>
> We hope we answered your questions, especially with regards to the experimental
> part. If so, we would kindly ask you to reconsider the rating of our paper. If
> you have any further doubts, we are looking forward to discussing further during
> the discussion period.

---

> > ### Comment · Reviewer_u8Z4 · 2024-08-14
> > **Feedback on rebuttal**
> >
> > I apologize for the late reply. I thank the authors for addressing my questions through clarifications and additional experiments. Accordingly, I have decided to raise my score to weak acceptance.

---

### Author Rebuttal · Authors · 2024-08-07

# Global response

We thank the area chair and the reviewers for their for their time spent reviewing our work, where we analyse the difficulty of learning from higher-order data correlations from the statistical point of view (via the likelihood ratio, LR), the computational point of view (via the low-degree LR), for random features (via replicas and simulations) and neural networks (via simulations).

We have replied to each of the reviewers directly; here we share a PDF with two additional figures:
- Prompted by a question from reviewer mvai we show the performance of exhaustive search on the spiked cumulant model to highlight the information-theoretic phase transition predicted by our theory in Fig 1.
- Prompted by a questions from reviewer u8Z4, we also repeated the experiments shown in Fig 1 and 2 with neural networks that were initialised such that all the neurons have small initial overlap with the spike.

We hope that our replies clarify all remaining doubts, and if not, look forward to discussing further during the revision period.

---

### Comment · Area_Chair_zNud · 2024-08-08

Dear authors, dear reviewers,

the discussion period has begun as the authors have provided their rebuttals.
I encourage the reviewers to read all the reviews and the corresponding rebuttals: the current period might be an opportunity for further clarification on the paper results and in general to engage in an open and constructive exchange.

Many thanks for your work.
The AC

---

### Decision · Program_Chairs · 2024-09-25

**Decision:**

Accept (poster)

**Comment:**

The submission focused on the problem of learning from higher-order cumulants (HOC), mainly by analyzing a spiked cumulant model introduced by Wang and Lu in 2017. In a classification task, the goal is to distinguish between two classes whose difference is the presence of a spike affecting the HOCs only, therefore the non-Gaussian features of the dataset. The authors provide theoretical bounds on the sample complexity for the distinguishability of the two clouds by means of a likelihood-ratio test. Moreover, they show that the task is efficiently achievable by a neural network, whereas random feature models cannot perform comparably in the same sample-complexity regime.

The paper obtained consensual positive reviews. Reviewers appreciated, in particular, its clarity and the offered insights on the sample complexity required by polynomial-time algorithms to learn HOCs by means of a simple, prototypical setup. Finally, a clear efficiency gap between random-feature models and neural networks is found. For this reason, I recommend acceptance of the manuscript.